# Tau activates microglia via the PQBP1-cGAS-STING pathway to promote brain inflammation

Meihua Jin [1,5], Hiroki Shiwaku [2,5], Hikari Tanaka [1,5], Takayuki Obita [3], Sakurako Ohuchi [3], Yuki Yoshioka [1], Xiaocen Jin [1], Kanoh Kondo [1], Kyota Fujita [1], Hidenori Homma [1], Kazuyuki Nakajima [4], Mineyuki Mizuguchi [3] & Hitoshi Okazawa [1✉]

Brain inflammation generally accompanies and accelerates neurodegeneration. Here we report a microglial mechanism in which polyglutamine binding protein 1 (PQBP1) senses extrinsic tau 3R/4R proteins by direct interaction and triggers an innate immune response by activating a cyclic GMP-AMP synthase (cGAS)-Stimulator of interferon genes (STING) pathway. Tamoxifen-inducible and microglia-specific depletion of PQBP1 in primary culture in vitro and mouse brain in vivo shows that PQBP1 is essential for sensing-tau to induce nuclear translocation of nuclear factor κB (NFκB), NFκB-dependent transcription of inflammation genes, brain inflammation in vivo, and eventually mouse cognitive impairment. Collectively, PQBP1 is an intracellular receptor in the cGAS-STING pathway not only for cDNA of human immunodeficiency virus (HIV) but also for the transmissible neurodegenerative disease protein tau. This study characterises a mechanism of brain inflammation that is common to virus infection and neurodegenerative disorders.

[1] Department of Neuropathology, Medical Research Institute and Center for Brain Integration Research, Tokyo Medical and Dental University, 1-5-45 Yushima, Bunkyo-ku, Tokyo 113-8510, Japan. [2] Department of Psychiatry, Graduate School of Medical and Dental Sciences, Tokyo Medical and Dental University, 1-5-45 Yushima, Bunkyo-ku, Tokyo 113-8510, Japan. [3] Faculty of Pharmaceutical Sciences, Graduate School of Innovative Life Science, University of Toyama, 2630 Sugitani, Toyama 930-0194, Japan. [4] Department of Bioinformatics, Institute of Bioinformatics, Soka University, 1-236 Tangi-machi, Hachioji, Tokyo 192-8577, Japan. [5] These authors contributed equally: Meihua Jin, Hiroki Shiwaku, Hikari Tanaka. ✉email: okazawa-tky@umin.ac.jp

Neurodegenerative diseases are associated with inflammation of the brain[1]. Extracellular disease proteins like Aβ[2] and damage-associated molecular patterns (DAMPs) such as high mobility group box protein 1 (HMGB1) or Aβ[3,4] released from sick neurons are the first group of stimuli for activating Toll-like receptor 2/4 (TLR2/4) or Receptor For Advanced Glycation End-Products (RAGE) on the membrane of microglia[5]. Molecules presented on the plasma membrane of dying neurons such as phosphatidyl serine (PS), Growth Arrest Specific 6 (GAS6), and Protein S are the second group of stimuli for activating dendritic cells including microglia and macrophages that possess PS receptors and TAM (Tyrosine-Protein Kinase Receptor TYRO3/AXL Receptor Tyrosine Kinase/Tyrosine-Protein Kinase Receptor Mer) receptors[6]. Intercellular Adhesion Molecule-3 (ICAM3), located on the plasma membrane of T cells, is activated by dying neurons[1] and is an indirect stimulus for dendritic cell-specific intercellular adhesion molecule-3-grabbing non-integrin (DC-SIGN) on microglia[7]. On the other hand, accumulating evidence indicate that inflammation accelerates neurodegenerative processes[1], suggesting a scheme of PCR-like amplification between degeneration and inflammation in the brains of neurodegenerative diseases.

Recent evidence from HIV infection sheds further light on the mechanism of microglia activation[8]. Reverse-transcribed single strand cDNA from the RNA genome of HIV interacts with PQBP1 as a sensor molecule, activates cGAS in their complex, and initiates transcription of inflammatory genes such as type-I interferons[8]. Subsequently, other proteins like Zinc Finger CCHC-Type Containing 3 (ZCCHC3)[9] and Ras-GTPase-Activating Protein SH3-Domain-Binding Protein 1 (G3BP1)[10] were identified as sensors for double-strand genome DNA of herpes simplex virus type 1 and vaccinia virus, or for DNA inducing Aicardi-Goutières syndrome, respectively, and the cGAS–STING pathway was established as the third paradigm for activating microglia in general except in neurodegenerative diseases.

Interestingly, PQBP1 was originally identified as a binding protein to polyglutamine (polyQ) tract amino acid sequence that is expanded in a group of neurodegenerative diseases called polyQ diseases[11–13]. PQBP1 possesses a folded WW domain[14–17] and an unfolded C-terminal domain categorized into intrinsically disordered proteins (IDPs)[18–22]. Loss of function of human PQBP1 by mutations causes intellectual disability and microcephaly[23–26]. PQBP1 functions as a regulator of RNA splicing, transcription, and DNA damage repair under physiological conditions[13,27–30] which is related mainly to neuronal functions[13,31–35], and possibly atherosclerosis[36]; however, it also functions as a sensor protein in macrophage and homologous cells of innate immune system for external molecules under certain pathological conditions[8].

The accumulated evidence bridging the cGAS–STING pathway, PQBP1, and neurodegenerative proteins provide us with an idea that PQBP1, possessing characteristics of IDP, might sense (or interact with) disease proteins that have an unfolded and denatured structure as extrinsic proteins, to drive the cGAS–STING pathway in the innate immune system especially in microglia[37]. Recent findings about prion-like propagation of disease proteins in the brains seem to support this idea[38–40].

Here, we show that tau protein interacts with PQBP1 in vitro and drives innate immune response of primary microglia in a PQBP1-dependent manner and that PQBP1 expression in microglia is important for tau-induced inflammation in the brain by using a conditional knockout mouse model of PQBP1 in microglia. In addition to the sensing mechanisms by cell surface receptors for extracellular and intercellular proteins, these results show a third mechanism of inflammation in the brain of

neurodegenerative diseases by an intracellular receptor, which could be generalized to multiple diseases since interaction rather than co-aggregation between the disease protein and PQBP1 occurs in Huntington's disease (HD) and spinocerebellar ataxia type 1 (SCA1)[12,13,41].

## Results

**Tau interacts with PQBP1 in microglia.** Tau protein has various alternative-splicing forms (Supplementary Fig. 1), among which we selected human Tau 410 (a representative 3 R tau) and Tau 441 (a representative 4 R tau) for interaction with human PQBP1. Human and also mouse PQBP1 proteins possess a WW domain (WWD), polar amino acid-rich domain (PRD) including dinucleotide repeats and C-terminal domain (CTD) with intrinsically disordered protein regions (IDP/IDRs) (Fig. 1a). We selected human PQBP1 full-length form (PQBP1-FL), human PQBP1(1–94) including WWD, and human PQBP1 (94–176) including PRD and human PQBP1 (193–265) including CTD. Surface plasmon resonance (SPR) analysis revealed that Tau 410 and Tau 441 full-length proteins almost equally interacted with WWD of PQBP1 (Fig. 1b and Supplementary Table 1).

To identify the binding region between tau and PQBP1, we performed interaction analysis using NMR spectroscopy. We measured 2D $^1$H-$^{15}$N HSQC NMR spectra of the full-length tau in the presence and absence of PQBP1. However, attempts to analyze the HSQC of full-length tau were complicated by extensive signal overlap and severe resonance line broadening. Therefore, we divided Tau 410 into five fragments (Fig. 1a) and analyzed the interaction of these fragments with PQBP1(1-94) (Fig. 1c). The addition of PQBP1(1–94) resulted in chemical shift changes and resonance line broadening in the HSQC spectra of Tau(151–197), Tau(198–243), and the repeat region Tau(244–341) (Fig. 1c). In particular, Tau(151–197) and Tau(198–243), which are proline-rich regions, showed considerable resonance line broadening (Fig. 1c). This result is consistent with binding of the proline-rich sequence to the WW domain of PQBP1 in our previous study[21]. On the other hand, no significant changes were observed in the HSQC spectra of Tau(1–150) and Tau(342–410) (Fig. 1c).

Given that these results suggested that PQBP1 selectively bound to the proline-rich regions, we further made point mutations (P179A and P216A) at two possible consensus sequences[15,16] for interaction with WW domain of PQBP1 (Fig. 1d), and investigated their affinities to PQBP1 by SPR (Fig. 1b). The results revealed obvious decreases in the affinity of single mutants (Tau410 P179A or Tau410 P216A) and a more remarkable decrease in the affinity of double mutant (Tau410 P179A/P216A) (Fig. 1b).

In addition, the interaction between PQBP1 and Tau 410 was confirmed by qualitative and quantitative immunoprecipitation assays (Fig. 1e). Moreover, in vivo interaction between PQBP1 and Tau was confirmed by immunoprecipitation and immunohistochemistry using cerebral cortex tissue from a mouse model (R6/2) of Huntington's disease (Supplementary Fig. 2) that could be considered as a tauopathy[42].

All of these results indicate that PQBP1 interacts with tau proteins by recognizing the proline-rich consensus sequences.

SPR analysis revealed that the interaction between PQBP1 and Aβ40 was weaker than that between PQBP1 and Tau 410/441 (Supplementary Fig. 3). Interaction between PQBP1 and α-synuclein was far weaker and not definitively determined (Supplementary Fig. 3).

**Dynamics of Tau interaction with PQBP1 in microglia.** Immunocytochemistry of primary mouse microglia revealed that

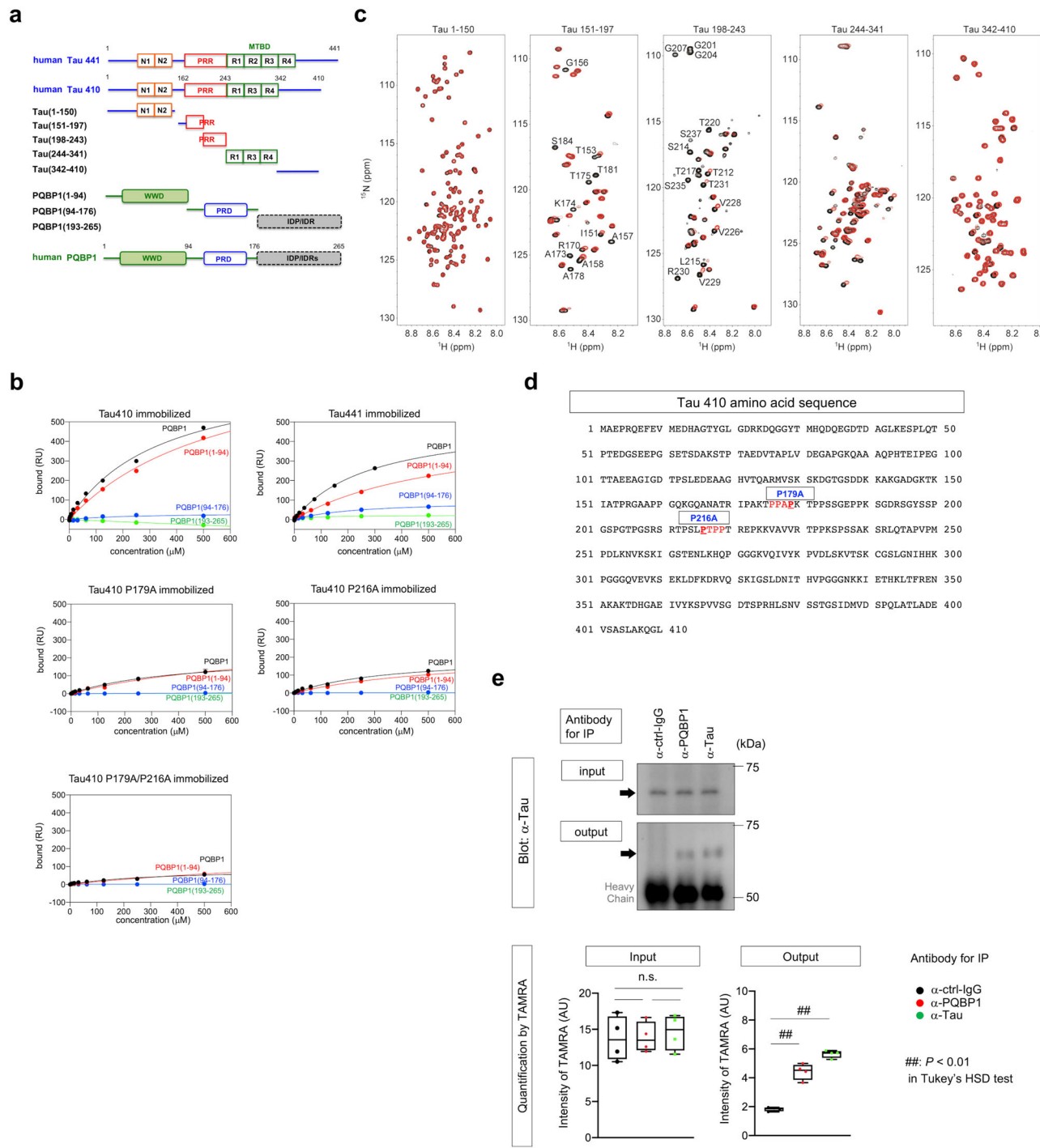

Tau 410/441 proteins were taken up from the culture medium into the cytoplasm of microglia and formed large cytoplasmic granules, where Tau 410/441 were colocalized with endogenous PQBP1 (Fig. 2a). Similar colocalization was observed between PQBP1 and the binding mutants of tau (Tau P179A, Tau P216A, or Tau P179A/P216A) at 48 h after the addition of tau to culture medium (Fig. 2a).

The cytoplasmic granules could be endosomes, phagosomes, endoplasmic reticuli (ER), or other cell organelles. Therefore, we prepared a soluble monomer of Tau 410/441 and their aggregated polymer after 10 days of incubation (Fig. 2b), and performed live imaging analysis of normal primary microglia that expressed PQBP1-EGFP by transfection of the expression vector and

incorporated four types of tau proteins from culture medium (Fig. 2c and Supplementary Movie 1).

First, PQBP1-EGFP was abundantly produced in rough ER surrounding the nucleus, and only a small portion was translocated into the nucleus (Fig. 2c and Supplementary Movie 1). At 0 h in the movie, Tau 410 or Tau 441 was added to the medium (Fig. 2c and Supplementary Movies 1 and 2, at 0 h). Tau incorporation into microglia became obvious after 4 h (Fig. 2c). Interestingly, tau rapidly translocated to the rough ER, presumably via endosome–ER contact sites[43–45]. The signals of PQBP1-EGFP were pushed out to smooth ER that is located at the peripheral side of the rough ER (Fig. 2c and Supplementary Movie 2, at 4 h), but then newly synthesized PQBP1-EGFP and tau were mixed up in rough ER (Fig. 2c and Supplementary

**Fig. 1 PQBP1 interacts with extrinsic Tau in microglia. a** Schematic presentation of deletion mutants of tau and PQBP1 used for SPR and NMR analyses. N1/2 N-terminal domain, R1/2/3/4 C-terminal microtubule-binding domain, PRR proline-rich region, WWD WW domain, PRD polar amino acid-rich domain, IDP/IDR intrinsically disordered region. **b** Equilibrium binding curves of PQBP1 to Tau 410, Tau 441, Tau 410 P179A, Tau 410 P216A, and Tau 410 P179A/P216A monitored by SPR. Tau proteins were immobilized on sensor chip surfaces. The binding response at equilibrium was plotted against the concentration of PQBP1. Black, full-length PQBP1; red, PQBP1(1–94); blue, PQBP1 (94–176); green, PQBP1 (193–265). RU resonance unit. **c** $^1H$-$^{15}N$ HSQC spectra of Tau 1–150, 151–197, 198–243, 244–341, and 342–410 in the absence (black) and presence (red) of PQBP1(1–94). Amino acid residue numbers of tau are based on Tau 410. The molar ratio of Tau:PQBP1 (1–94) is 1:5. Residues showing significant chemical shift perturbation and line broadening of resonances are labeled in the HSQC spectra of Tau 151–197 and 198–243. Ctrl control, ppm parts per million. **d** Tau 410 amino acid sequence showing consensus motifs for interaction with PQBP1-WWD (red letter) and mutated prolines (underlined). **e** A representative western blot showing co-precipitation of tau with PQBP1 from a lysate of C57BL/6 microglia cultured for 3 days with 25 nM non-aggregated Tau-TAMRA (upper panels). Quantitative immunoprecipitation was performed with C57BL/6 microglia cultured for 3 days with 25 nM non-aggregated Tau-TAMRA. Input was the intensity of TAMRA-labeled Tau 410 in the C57BL/6 microglia lysate samples used for immunoprecipitation. Output was the intensity of TAMRA-labeled Tau 410 in samples after immunoprecipitation with anti-tau, anti-PQBP1, and human IgG. The fluorescence intensity of TAMRA-labeled Tau 410 was measured on a FLUOstar OPTIMA-6 microplate reader. n.s. not significant, AU arbitrary unit. $N = 4$ per group. $P = 0.9963$ (ctrl-IgG vs PQBP1), 0.8874 (ctrl-IgG vs tau), 0.9215 (PQBP1 vs tau), $9.63e^{-6}$ (PQBP1 vs ctrl-IgG), $3.69e^{-7}$ (tau vs ctrl-IgG). $^{##}P < 0.01$, $^{n.s.}P > 0.05$ in Tukey's HSD test. Box plots show the median, quartiles, and whiskers that represent data outside the 25th to 75th percentile range.

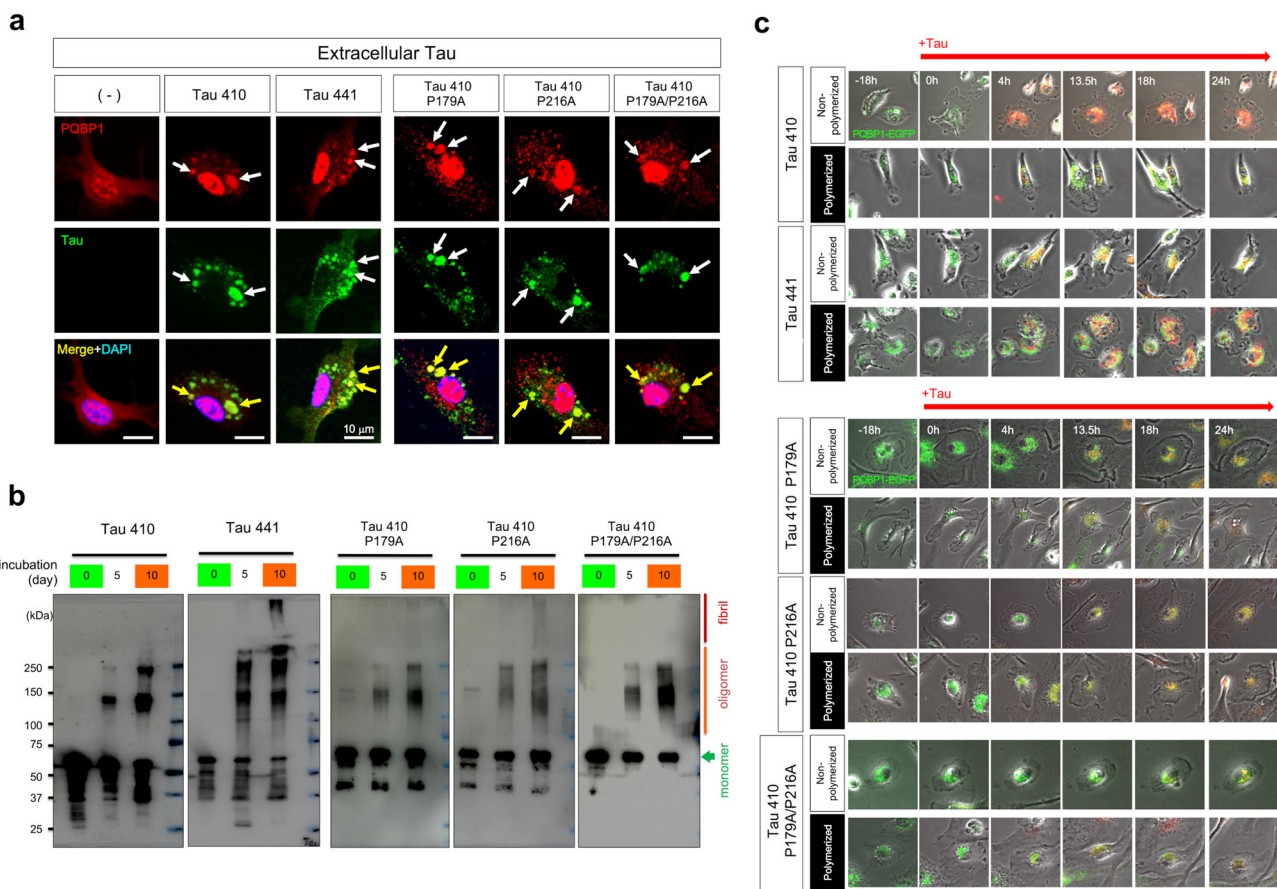

**Fig. 2 Colocalization of PQBP1 and incorporated Tau in microglia. a** Immunocytochemistry of primary microglia prepared from C57BL/6 mice cultured with Tau 410, Tau 441, Tau 410 P179A, Tau 410 P216A, or Tau 410 P179A/P216A proteins in the medium. Colocalization of PQBP1 and extrinsic tau was detected frequently. The experiments were repeated four times independently with similar results. **b** In vitro polymerization of Tau 410, Tau 441, Tau 410 P179A, Tau 410 P216A, and Tau 410 P179A/P216A full-length proteins. Only Tau 441 and Tau 410 P216A partially formed fibrils after 10 days of incubation, while the other full-length protein did not. The experiments were repeated four times independently with similar results. **c** Images of normal microglia before and after addition of polymerized or non-polymerized Tau 410, Tau 441, Tau 410 P179A, Tau 410 P216A, or Tau 410 P179A/P216A, among which three species are shown in Supplementary Movies 1–3. h hour.

Movie 2, after 12 h). Thereafter, the signals of PQBP1–Tau complex diffusely leaked from ER to the cytosol (Fig. 2c and Supplementary Movie 2, after 13 h).

Basically, Tau 410 and Tau 441 behaved similarly (Fig. 2c and Supplementary Movie 1). However, tau and PQBP1 signals merged immediately in ER in the case of Tau 410 monomer, while the merge of signals was delayed in the case of polymerized

Tau 441 (Fig. 2c and Supplementary Movies 1 and 2). Interestingly, STING is located in ER (subcellular location in UniProt https://www.uniprot.org/uniprot/Q86WV6) while cGAS is located in cytosol (subcellular location in UniProt https://www.uniprot.org/uniprot/Q8N884) after stimulation. Therefore, the Tau–PQBP1–cGAS–STING complex could be formed in the prograde order of the dynamics of tau observed above (Fig. 2c

and Supplementary Movie 1), or the Tau–PQBP1–cGAS–STING complex might be formed in retrograde order[8] during the reflux from cytosol to ER that may occur under the collapse of the ER–cytosol border (Fig. 2c and Supplementary Movie 2).

In the case of binding mutants of tau (Tau410 P179A, Tau410 P216A, Tau410 P179A/P216A) in monomer and polymerized states, live imaging analysis of normal primary microglia suggested a tendency for delayed merging between PQBP1 and the binding mutants of Tau in ER (Fig. 2c and Supplementary Movie 3).

It is very interesting that direct incorporation of extrinsic tau to rough ER is quite homologous to the direct incorporation of coronavirus RNAs causing COVID-19, MERS, and SARS into rough ER for the formation of double membrane vesicle (DMV)[46,47].

**Tau activates cGAS–STING pathway in microglia**. To examine whether extrinsic tau proteins activate cGAS–STING pathway in microglia, we first confirmed nuclear translocation of nuclear factor κB (NFκB) (Fig. 3a) at the final step of the cGAS–STING pathway[8,48]. As expected, we observed nuclear translocation of NFκB-p65 in more than 80% of microglia by the addition of Tau 410/441 proteins to the culture medium (Fig. 3a). Meanwhile, addition of binding mutants of Tau did not induce nuclear translocation of NFκB so efficiently (Fig. 3a). Moreover, polymerized tau proteins, after incubation for 10 days, induced nuclear translocation of NFκB less efficiently than the monomer state of tau (no polymerization) (Fig. 3a). We calculated the ratio of nuclear NFκB-positive microglia in all cases (Fig. 3b).

A similar difference was observed in the induction of *Tnf* and *Isg54* gene expression in microglia (Fig. 3c). Tau 410 and Tau 441 isoforms did not show obvious differences in inducing nuclear translocation of NFκB and *Isg54* gene expression (Fig. 3b, c), while Tau 441 monomer and polymer did not efficiently induce *Tnf* gene expression (Fig. 3c). The binding mutants of tau did not efficiently induce *Tnf* and *Isg54* gene expression in microglia (Fig. 3c).

The independency of human Tau 410 and Tau 441 isoforms in activating the cGAS–STING pathway in mouse microglia could be a common mechanism across various tauopathies including Alzheimer's disease (AD), given that different composition of 4R/3R tau isoforms recapitulate progression patterns of the corresponding human diseases independently of tau species and mouse strains[49]. Tau-induced activation of cGAS–STING pathway might underlie a more general inflammatory response that is independent of progression patterns of tauopathy in the central nervous system.

**PQBP1 is essential for tau-induced cGAS–STING activation in microglia**. Combination of these two lines of results prompted us to test further whether PQBP1 is essential for tau-induced cGAS–STING activation. We generated microglia-specific and tamoxifen-inducible PQBP1 conditional knockout mouse (*Pqbp1*-cKO male: $Cx3cr1^{CreER/CreER}/Pqbp1^{floxX/Y}$, female: $Cx3cr1^{CreER/CreER}/Pqbp1^{floxX/floxX}$) by crossing *Pqbp1*-floxed mice (Supplementary Fig. 4a) and tamoxifen-inducible $Cx3cr1^{CreER}$ mice (Supplementary Fig. 4b), in which tamoxifen-inducible depletion of PQBP1 in microglia was confirmed by immunohistochemistry of cerebral cortex tissues (Supplementary Figs. 5 and 6).

Primary microglia prepared from newborn homozygous *Pqbp1*-cKO mice were stimulated by addition of Tau 410/441 in the culture medium (Fig. 4a). While Tau 410 and Tau 441 induced nuclear translocation of NFκB in all cells of microglia prepared from *Pqbp1*-cKO mice in the absence of tamoxifen, tau

proteins induced nuclear translocation of NFκB only in one half of microglia in the presence of tamoxifen (Fig. 4b). The inhibitory effect on nuclear translocation of NFκB was similarly observed in microglia transfected with siRNA of cGAS or STING (Fig. 4b). Quantification of the number of cells with nuclear translocation of NFκB (p65) supported this conclusion (Fig. 4c). Western blot revealed an increase in the active form of NFκB (pSer536-NFκB-p65) by Tau 410/441 in normal microglia (Fig. 4d). Moreover, western blot analysis with an antibody against the active form of IRF3 (pSer396-IRF3) revealed that IRF3 activation, another downstream transcription factor of the cGAS–STING pathway[50] (Fig. 4d). However, such activation of NFκB and IRF3 were suppressed by PQBP1 depletion, knockdown of cGAS or knockdown of STING in microglia (Fig. 4d).

It is noteworthy that depletion of PQBP1 and suppression of cGAS or STING reduced nuclear translocation of NFκB to nearly 50% but not completely, suggesting the existence of pathway(s) other than the PQBP1–cGAS–STING pathway that could induce NFκB activation.

Other functions of PQBP1-depleted microglia, such as chemotaxis (Fig. 4e) and phagocytosis (Fig. 4f), were unchanged, while it remains possible that other functions of microglia may be affected by depletion of PQBP1, given that PQBP1 is involved in transcriptional[12–14] and post-transcriptional gene expression[26–30,34,35].

**PQBP1 is essential for cGAS to recognize Tau and for microglia activation**. Intracellular dynamics and localization of multiple cGAS–STING complex components have not been elucidated in detail. However, physical colocalization of tau and cGAS is essential for recognition of target proteins coming from the extracellular space. We found in wild-type primary microglia that Tau 410/441 monomer proteins taken up from the culture medium, colocalized with cGAS after AT8 antibody-recognizable phosphorylation (Fig. 5a). Meanwhile, in *Pqbp1*-cKO microglia, cGAS protein was not recruited to the foci of phosphorylated tau proteins (Fig. 5a). The tendency was confirmed by quantification of the ratio of Tau-cGAS colocalized microglia to all microglia (Fig. 5b). The result indicated that PQBP1 guides cGAS proteins to tau protein foci in microglia.

Next, we tested the effect of PQBP1 on transcriptional function of NFκB under the stimulation of Tau 410/441 by performing a luciferase assay for the activity of NFκB cis-element and quantitative reverse transcription polymerase chain reaction (RT-qPCR) for expression of target genes of NFκB (Fig. 5c). Luciferase assay using the reporter plasmid containing the consensus *cis*-element for NFκB was activated by Tau 410/441 stimulation, and revealed that partial deficiency of PQBP1, cGAS, and STING suppressed the activation of NFκB *cis*-element (Fig. 5d). Internal control (renilla) was used for correcting the specific signal of firefly in the luciferase assays (Fig. 5d). *Pqbp1*-cKO microglia and siRNA-transfected microglia were used to prove the essential function of PQBP1, cGAS, and STING for activation of NFκB *cis*-element (Fig. 5d). RT-qPCR also revealed that Tau 410/441 stimulation transcriptionally upregulated target genes of NFκB, such as *Tnf*, *Isg54*, *Cxcl10*, and *Ifnβ*, indicating that PQBP1, cGAS, and STING were essential for the upregulation (Fig. 5e).

**PQBP1 depletion in microglia rescues tau-induced brain inflammation**. Finally, we examined whether PQBP1 intracellular receptor senses extrinsic tau and activates the cGAS–STING pathway in mouse brains in vivo. For this purpose, we prepared tamoxifen-inducible microglia-specific homozygous *Pqbp1*-cKO mice (Supplementary Fig. 4), in which PQBP1 is completely

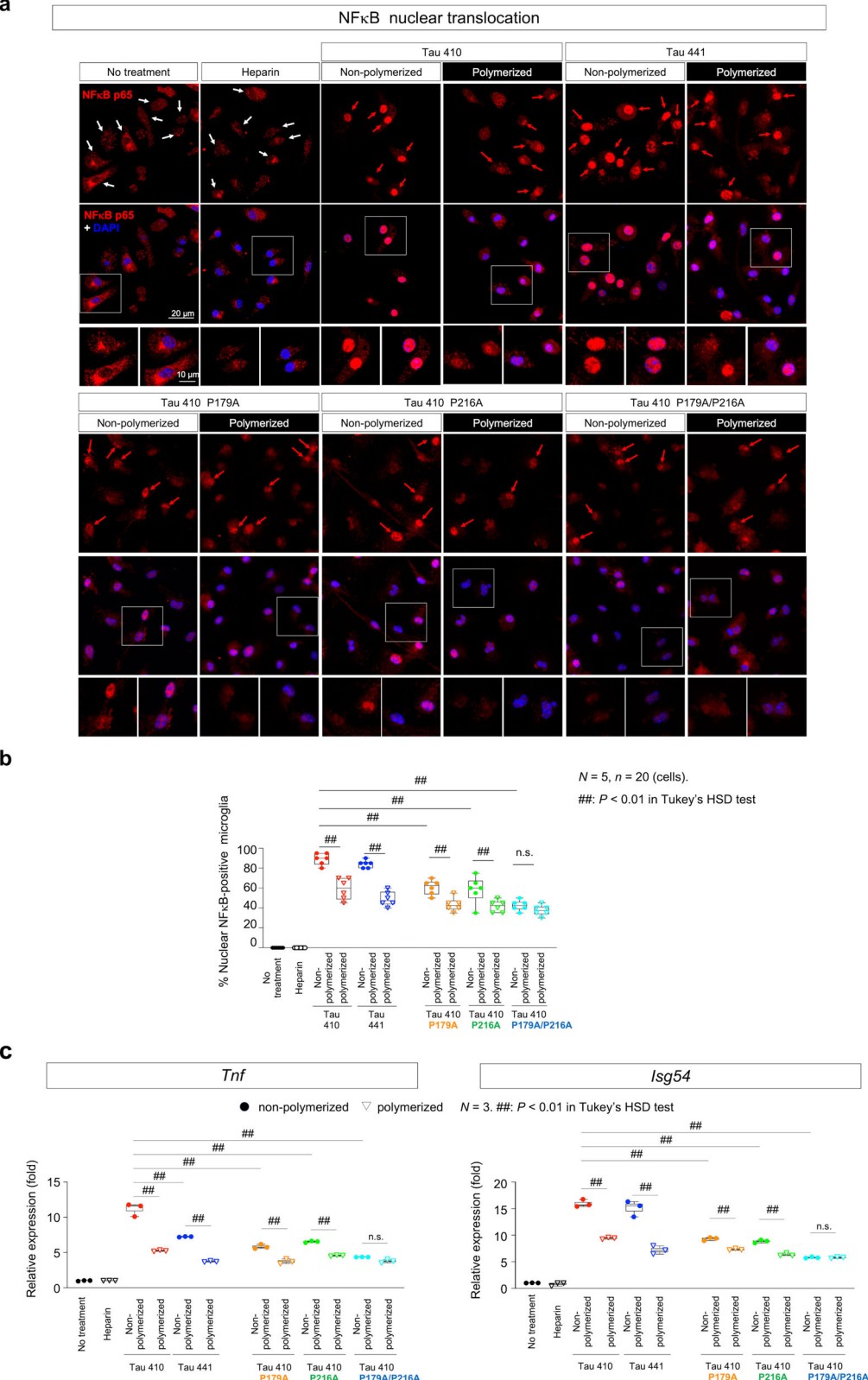

**a** NFκB nuclear translocation

**b** $N = 5$, $n = 20$ (cells).
##: $P < 0.01$ in Tukey's HSD test

**c** _Tnf_ _Isg54_
● non-polymerized △ polymerized $N = 3$. ##: $P < 0.01$ in Tukey's HSD test

depleted in microglia (Supplementary Figs. 5 and 6). Tamoxifen by itself was not so toxic to influence the number of microglia at the dose used to induce Cre (Supplementary Fig. 6, Iba1 staining).

We prepared four groups of mice: _Pqbp1_-cKO injected with Tau 410 monomer after tamoxifen treatment, _Pqbp1_-cKO injected with phosphate-buffered saline (PBS) after tamoxifen treatment, _Pqbp1_-cKO injected with Tau 410 monomer after

mock treatment, and _Pqbp1_-cKO injected with PBS after mock treatment. We injected Tau 410 monomer or a similar volume of solvent to bilateral entorhinal cortex (2.2 μg/μl, 1 μl/injection, 4.4 μg/mouse) (Fig. 6a).

Immunohistochemistry of control group brains (non-induced _Pqbp1_-cKO injected with Tau 410 monomer) revealed tau incorporation, Tau-PQBP1 colocalization, nuclear translocation

**Fig. 3 Extrinsic Tau proteins induce nuclear translocation of NFκB and expression of pro-inflammatory genes in microglia. a** Immunocytochemistry of primary microglia prepared from newborn C57BL/6 mice reveals nuclear translocation of NFκB at 48 h after addition of recombinant full-length Tau 410, Tau 441, Tau 410 P179A, Tau 410 P216A, or Tau 410 P179A/P216A protein in the non-polymerized or polymerized state to culture medium. Images were obtained by confocal microscopy. Red or white arrows indicate microglia with or without nuclear translocation of NFκB. Lower panels show high magnification of representative cells. In the presence of tau proteins in culture medium, nuclear translocation of NFκB was induced in most microglia, while in the absence of tau proteins NFκB remained in the cytoplasm. **b** The effects of various tau species in the polymerized and non-polymerized state on nuclear translocation of NFκB were compared. Aggregation of Tau 410/441 reduces the frequency of NFκB nuclear translocation. PQBP1-binding proline single mutants (Tau 410 P179A, Tau 410 P216A) partially, while double proline mutant (Tau 410 P179A/P216A) substantially, lost the ability to induce NFκB nuclear translocation. $N = 5$, $n = 20$ (cells). ##$P < 0.01$ in Tukey's HSD test. **c** Isoforms, mutations, and aggregation states of tau affect induction of *Tnf* and *Isg54* gene expression. Consistent with immunocytochemistry of NFκB, non-polymerized Tau 410 induced pro-inflammatory genes most efficiently, while Tau 441 possesses 60–80% of the induction ability. PQBP1-binding proline single mutants (Tau 410 P179A, Tau 410 P216A) partially, while double proline mutant (Tau 410 P179A/P216A) substantially, lost transactivation of *Tnf* and *Isg54* gene expression. RT-qPCR reverse transcriptase quantitative polymerase chain reaction. $N = 3$. ##$P < 0.01$ in Tukey's HSD test. Box plots show the median, quartiles, and whiskers that represent data outside the 25th to 75th percentile range.

of NFκB (p65), and cGAS recruitment to the complex were observed in microglia gathering to injected area of entorhinal cortex (Fig. 6b, left panels). On the other hand, PQBP1 colocalization to tau, nuclear translocation of NFκB or cGAS recruitment to the complex was not found in microglia of tamoxifen-induced *Pqbp1*-cKO mice (Fig. 6b, middle panels). Quantification of such changes in microglia confirmed these conclusions (Fig. 6b, right graphs). Microglia activation with nuclear translocation of NFκB occurred at a high frequency (89%) when tau was incorporated (Fig. 6b, right graphs), while the ratio was low (6.8%) among total microglia (Tau-positive and Tau-negative microglia) (Supplementary Fig. 7) because tau incorporation was not so efficient.

The similarity to the disease-specific microglia[51] to microglia activated in tau was also checked by using anti-lipoprotein lipase (LPL) antibody (Supplementary Fig. 8). We observed LPL-positive microglia similar to the disease-specific microglia reported previously[51], and when tau was injected into mouse brains, their total number decreased in the absence of PQBP1 according to total number of microglia (Supplementary Fig. 8), while the ratio of LPL-positive microglia among total microglia was unchanged (Supplementary Fig. 8). LPL in microglia is essential for its lipid uptake and regulates immune reactivity of ordinary microglia[52]. LPL immunoreactivity is linked to amyloid plaques while it is not associated with neurofibrillary tangles[53]. These notions might explain why tau injection did not induce a disproportionate increase of LPL-positive microglia.

RT-qPCR of *Tnf* and *Isg54* with entorhinal cortex reconfirmed enhanced brain inflammation in the control mouse group (*Pqbp1*-cKO after mock treatment injected with Tau 410 monomer) while it was recovered by microglia-specific depletion of PQBP1 (Fig. 6c). Y-maze test and Morris water maze test supported that microglia-specific depletion of PQBP1 recovered cognitive symptoms based on the extracellular tau-induced brain inflammation (Fig. 6d).

In vivo live imaging of microglia by two-photon microscopy with B6.129P2(Cg)-*Cx3cr1*[tm1Litt]/J (*CX₃CR-1*[GFP]) mice in which microglia were labeled by endogenous *Cx3cr1* enhancer/promoter-driven EGFP protein revealed that microglia incorporated red particles of TAMRA-NHS-labeled Tau 410 full-length protein injected in monomer state (Fig. 6e). A part of such microglia (yellow cell body) with Tau 410 signals became active in movement of branches and enlarged into cell bodies (Fig. 6e, upper panels). Such activation of microglia by tau protein was suppressed by co-injection of siRNA against PQBP1 (Fig. 6e, lower panels) that efficiently suppressed PQBP1 (Fig. 6f).

**PQBP1-binding mutations of Tau reduce inflammatory responses of microglia in brain.** Inflammatory responses of

microglia to tau proteins were also reduced in vivo when they were mutated at the binding motifs to PQBP1 (Fig. 7), similar to the case in vitro (Fig. 3). We injected normal Tau 410 or the binding mutants of tau (Tau410 P179A, Tau410 P216A, or Tau410 P179A/P216A) to bilateral entorhinal cortex of non-induced or tamoxifen-induced microglia-specific homozygous *Pqbp1*-cKO mice (Fig. 7a). Colocalization of tau proteins to PQBP1 (Fig. 7b) was abolished in tamoxifen-induced *Pqbp1*-cKO mice (Fig. 7c). Recruitment of cGAS was reduced to PQBP1-Tau-double-positive foci, when tau binding mutants were injected (Fig. 7b), while the recruitment of cGAS was completely abolished in tamoxifen-induced *Pqbp1*-cKO mice (Fig. 7c). The final outcome, nuclear translocation of NFκB, was decreased in the case of tau-binding mutants (Fig. 7b), whereas it was not completely abolished in tamoxifen-induced *Pqbp1*-cKO mice (Fig. 7c), suggesting the existence of another pathway inducing NFκB activation, in addition to the pathway based on the tau–PQBP1–cGAS complex.

Quantitative RT-PCR of *Tnf* and *Isg54* with brain tissues of Tau-injected area reconfirmed the reduced inflammatory responses of microglia to tau-binding mutants (Fig. 7d). The residual inductions of *Tnf* and *Isg54* by Tau410 P179A or Tau410 P216A were further suppressed by depletion of PQBP1 in tamoxifen-induced microglia-specific *Pqbp1*-cKO mice (Fig. 7e).

**Relationship between PQBP1 and LRP1 or TREM2.** We addressed the mechanistic relationship of PQBP1 to the molecules playing critical roles in Tau-mediated pathology, low-density lipoprotein receptor-related protein 1 (LRP1) and triggering receptor expressed on myeloid cells 2 (TREM2). LRP1 is located at Clathrin-coated pit of cell membranes, and incorporates relatively large extracellular molecules, including lipoproteins, proteinase and inhibitor complexes, matrix proteins and other proteins into the cell via endosomes, and functions in macrophage for atherosclerosis[54]. LRP1 is also expressed in microglia[55] and regulates the microglial immune response[56,57]. LRP1 was recently demonstrated as a dominant tau receptor[58]. TREM2 is a membrane receptor coupled with DNAX-activating protein (12 kDa; DAP12) that transduces various signals including ERK and PI3K that activate NFκB[59,60]. It has been identified as a risk factor for AD[61,62], and is expressed in microglia to impede tau seeding and tau pathology in mouse models[63–65].

We transfected the double nickase plasmid that expressed green fluorescent protein (GFP) for identification of transfected cells (Fig. 8a, left images) and almost completely knocked out *Lrp1* or *Trem2* gene (Fig. 8a, right graphs) in microglia prepared from tamoxifen-inducible microglia-specific *Pqbp1*-cKO mice. Control plasmid expressed scrambled 20 nt gRNA. NFκB activity was monitored by two reporter plasmids expressing red

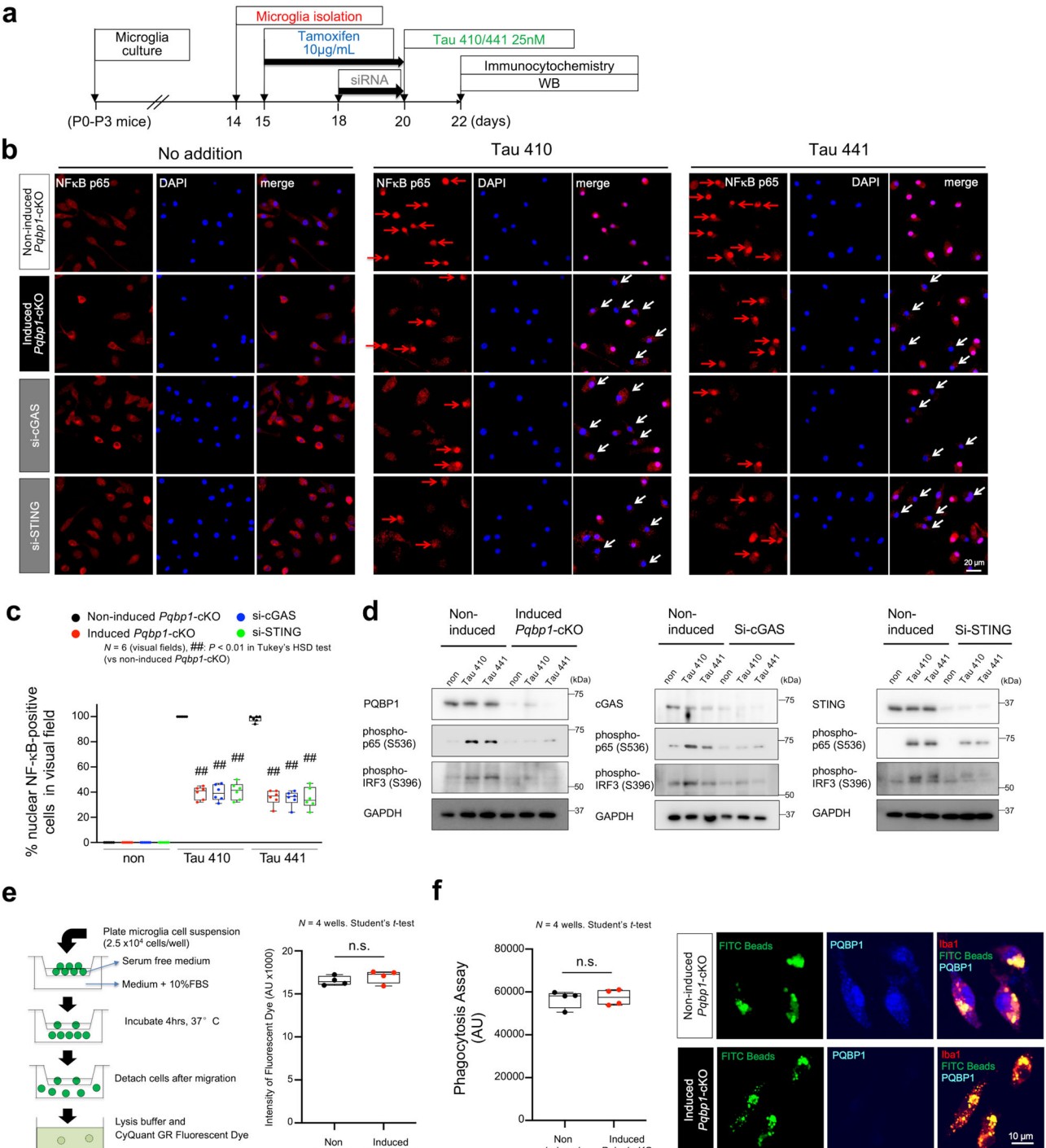

**Fig. 4 PQBP1–cGAS–STING pathway mediates Tau-induced NFκB activation in microglia. a** Protocol of tamoxifen-induced *Pqbp1* knockout in primary microglia prepared from newborn *Pqbp1*-cKO mice. WB western blot. **b** Effect of PQBP1 deficiency, cGAS knockdown, and STING knockdown on Tau 410/ 441-induced nuclear translocation of NFκB in microglia. Activated microglia with nuclear NFκB are indicated with red arrows. Non-activated microglia are indicated with white arrows. Images were obtained by confocal microscopy. **c** Quantitative analyses of % nuclear NFκB-positive microglia in six visual fields in normal condition, PQBP1 deficiency, cGAS knockdown, and STING knockdown. $N = 6$ (visual fields). $^{##}P < 0.01$ in Tukey's HSD test (vs non-induced *Pqbp1*-cKO). **d** Western blots reveal activation of NFκB (pSer536-p65) and IRF3 (pSer396-IRF3) after culture with non-polymerized Tau 410/441 in normal microglia, but not in tamoxifen-induced *Pqbp1*-cKO, siRNA-mediated cGAS-KD, or siRNA-mediated STING-KD microglia. This experiment was repeated independently, three times, with similar results. **e** Effect of PQBP1 depletion on microglia migration. FBS fetal bovine serum. $N = 4$ wells. Statistical test: Student's *t*-test (two-sided). **f** Effect of PQBP1 depletion on microglia phagocytosis by using FITC-beads. AU arbitrary unit. $N = 4$ wells. For statistical analysis, a Student's *t*-test (two-sided) was performed. Box plots show the median, quartiles, and whiskers that represent data outside the 25th to 75th percentile range.

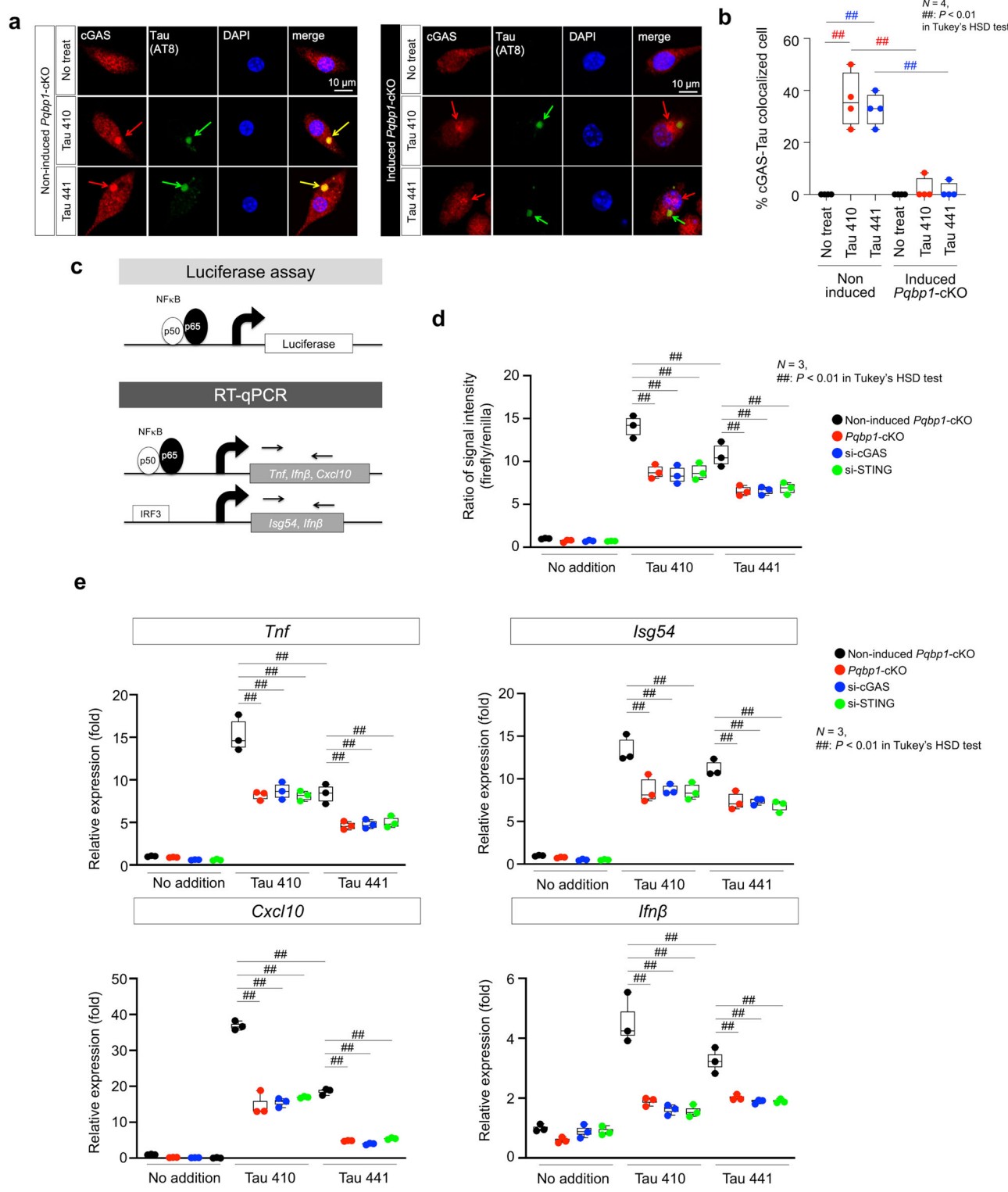

fluorescent protein (RFP) or luciferase driven by NFκB promoter (Fig. 8b). NFκB activation in microglia by addition of Tau 410 to the culture medium was suppressed by *Lrp1*-KO or *Trem2*-KO in the RFP signal assay, as expected (Fig. 8c). In the case of additional *Pqbp1*-cKO by tamoxifen, NFκB activity was unchanged in *Lrp1*-KO microglia, but further suppressed in *Trem2*-KO (Fig. 8c). The discrepancy between NFκB activities in *Lrp1*-KO and *Trem2*-KO cells under the additional *Pqbp1*-cKO was also confirmed by luciferase assay (Fig. 8d). These genetic

analyses indicated that PQBP1 and LRP1 are in the same pathway, while PQBP1 is independent of TREM2 (Fig. 8e).

**Direct and indirect effects of Tau on neuronal death**. MAP2 immunohistochemistry of Tau 410-injected mice revealed that non-microglia cells incorporating tau (Fig. 6b, middle panels) were mostly neurons (Fig. 9a). This finding, together with accumulated knowledge of tau propagation[66], prompted us to comprehensively address relationship of the direct effect of tau incorporated into neurons and the indirect effect of tau-induced

**Fig. 5 PQBP1 is essential for recruiting cGAS to Tau.** Confocal microscopy images of primary *Pqbp1*-cKO microglia stained with anti-cGAS and -tau antibodies. **a** Cytoplasmic tau foci after incorporation recruit cGAS in the presence of PQBP1 (left panels), while in the absence of PQBP1, cGAS is mislocalized and not merged with cytoplasmic foci of tau, which is phosphorylated after endocytosis. **b** Quantification of cGAS-tau colocalized cells in four wells of non-induced and tamoxifen-induced *Pqbp1*-cKO microglia after tau addition to primary culture medium. $N = 4$. $^{##}P < 0.01$ in Tukey's HSD test. **c** Schemes showing the concepts of the luciferase assay used to monitor NFκB activity under tau stimulation and PQBP1–cGAS–STING pathway perturbation (**d**) and of RT-qPCR used to monitor expression levels of NFκB-target genes (**e**). **d** Luciferase assay to monitor NFκB activity under Tau 410/441 stimulation together with PQBP1–cGAS–STING pathway perturbation by tamoxifen-induced *Pqbp1*-cKO, siRNA against cGAS and siRNA against STING. $N = 3$. $^{##}P < 0.01$ in Tukey's HSD test. **e** RT-qPCR to monitor expression levels of NFκB- or IRF3-target genes (*Tnf, Isg54, Cxcl10,* and *Ifnβ*) under Tau 410/441 stimulation together with PQBP1–cGAS–STING pathway perturbation by tamoxifen-induced *Pqbp1*-cKO, siRNA against cGAS, and siRNA against STING. $N = 3$. $^{##}P < 0.01$ in Tukey's HSD test. Box plots show the median, quartiles, and whiskers that represent data outside the 25th to 75th percentile range.

microglia activation, especially on neuronal death (Fig. 9b). Terminal deoxynucleotidyl transferase dUTP nick end labeling (TUNEL) assay revealed four groups of neurons from the aspect of tau incorporation and neuronal death (Fig. 9b). In non-polymerized Tau 410 injection, cell death was significantly increased in neurons that did not incorporate injected tau (Tau-negative neurons); this increase was suppressed by PQBP1-binding mutations of tau (Fig. 9b, left graphs). Consistently, the increase of cell death in Tau-negative neurons was suppressed by tamoxifen-induced *Pqbp1*-cKO (Fig. 9b, first line right graph). The effect of Tau 410 on Tau-negative neurons was also suppressed by tau polymerization (Fig. 9b, first line left graph), consistent with the decreased inflammatory gene responses of microglia by tau polymerization (Fig. 3). On the other hand, cell death was increased by tau polymerization in neurons that incorporated tau (Tau-positive neurons), which was not suppressed by PQBP1-binding mutations of tau (Fig. 9b, left graphs) or by PQBP1 depletion (Fig. 9b, right graphs).

In contrast to the in vivo effect in the brain where neurons and microglia co-existed, Tau 410 addition to primary neuron culture revealed that non-polymerized Tau 410 induced no neuronal death in the absence of microglia (Fig. 9c). Meanwhile, neuronal cell death tended to increase following the incorporation of polymerized Tau 410 but not significantly (Fig. 9c). All of these results suggested that tau monomer-induced and PQBP1-dependent activation of microglia indirectly induced cell death of Tau-negative neurons, and that incorporation of polymerized tau to neurons directly induced cell death to a lesser extent (Fig. 9d). The indirect pathway via microglia is mediated by cytokines such as TNF, ISG54, CXCL10, and IFNβ in LPL-negative microglia. Meanwhile, LPL-positive microglia around Aβ plaques might inhibit tau seeding[63], and LPL-negative microglia in the absence of Aβ plaques might somehow contribute to tau aggregate seeding[67].

**FTLD-linked disease mutation of Tau does not affect PQBP1-dependent microglial activation.** To elucidate the pathological meaning of FTLD-linked disease mutation of tau (P301S)[68,69] in PQBP1-mediated microglia activation, we compared Tau 441 and Tau 441 P301S regarding affinity to PQBP1, induction of nuclear translocation of NFκB, gene expression of inflammation genes, and direct effect on cell death in primary neuron culture (Supplementary Fig. 9). Their affinities to PQBP1 were equivalent in SPR (Supplementary Fig. 9a and Supplementary Table 2); they were similarly taken up by primary microglia to merge with PQBP1 at cytoplasmic foci (Supplementary Fig. 9b); their dynamics for merging with PQBP1 were similar even if they were polymerized (Supplementary Fig. 9c, d); they similarly induced nuclear translocation of NFκB in primary culture microglia, which were similarly decreased by polymerization (Supplementary Fig. 9e); they similarly upregulated *Tnf* and *Isg54* gene expression in RT-PCR of primary microglia, which were similarly

suppressed by polymerization (Supplementary Fig. 9f); in immunohistochemistry of mouse brain, they were similarly colocalized with PQBP1 and cGAS at cytoplasmic foci of microglia (Supplementary Fig. 9g); and they similarly induced nuclear translocation of NFκB in brain microglia, which was equally suppressed by tamoxifen-induced *Pqbp1*-cKO (Supplementary Fig. 9g). Although these data indicated similar PQBP1-dependent microglial activation by Tau 441 and Tau 441 P301S, induction of pro-inflammatory genes was more prominent by Tau 441 P301S than Tau 441 (Supplementary Fig. 9h).

TUNEL staining of the injected mice revealed enhancement of neuronal cell death by Tau P301S mutation and polymerization (Supplementary Fig. 9i), which was independent of PQBP1 deficiency in microglia of tamoxifen-induced *Pqbp1*-cKO (Supplementary Fig. 9i). As expected, Tau P301S incorporation directly induced such neuronal cell death in primary neurons without microglia (Supplementary Fig. 9j).

The discrepancy between the effects of Tau 441 P301S on microglia in vitro and in vivo for induction of pro-inflammatory genes could be explained by non-cell autonomous factors originated from dying neurons that should release various factors including DAMPs like HMGB1 and Aβ[4,70–72] and senescence-associated secretary phenotype factors (SASPs)[73].

**PQBP1 in microglia as a possible therapeutic target.** Finally, we examined how normal form, binding mutants and disease mutant of tau in non-polymerized or polymerized state affect mouse memory functions after their injection into bilateral entorhinal cortex (Fig. 7a) by using the Y-maze test for short-term memory and the Morris water maze test for long-term memory (Fig. 10a). In non-polymerized state, both tests revealed that Tau 410 strongly, but Tau 441 and Tau 441 P301S weakly, impaired memory functions (Fig. 10a, upper graphs). The impairment was recovered when Tau 410 possessed mutations for binding to PQBP1 (Fig. 10a, upper graphs). Impairment of memory by Tau 410 and Tau 441 was also recovered when PQBP1 was depleted in microglia, while the effect of Tau 441 P301S was resistant to microglial PQBP1 depletion (Fig. 10a, upper graphs). These results of memory functions were consistent with the in vivo effects of tau species on neuronal death and their modification by microglial PQBP1 depletion (Fig. 10b and Supplementary Fig. 9i).

On the other hand, when tau species were polymerized (Fig. 10a, lower graphs), Tau 441 impaired memory functions especially when it had the disease-linked mutation (Fig. 10a, lower graphs), though Tau 410 and their PQBP1-binding mutants did not affect memory function (Fig. 10a, lower graphs). Together with their effects on neuronal cell death in vivo (Fig. 9b, Supplementary Fig. 9i) and their effect on neuronal cell death in vitro (Fig. 9c, Supplementary Fig. 9j), the memory phenotypes could be explained by direct toxicities of Tau 441 and Tau 441 P301S taken up by neurons that were higher than Tau 410 or the PQBP1-binding mutants.

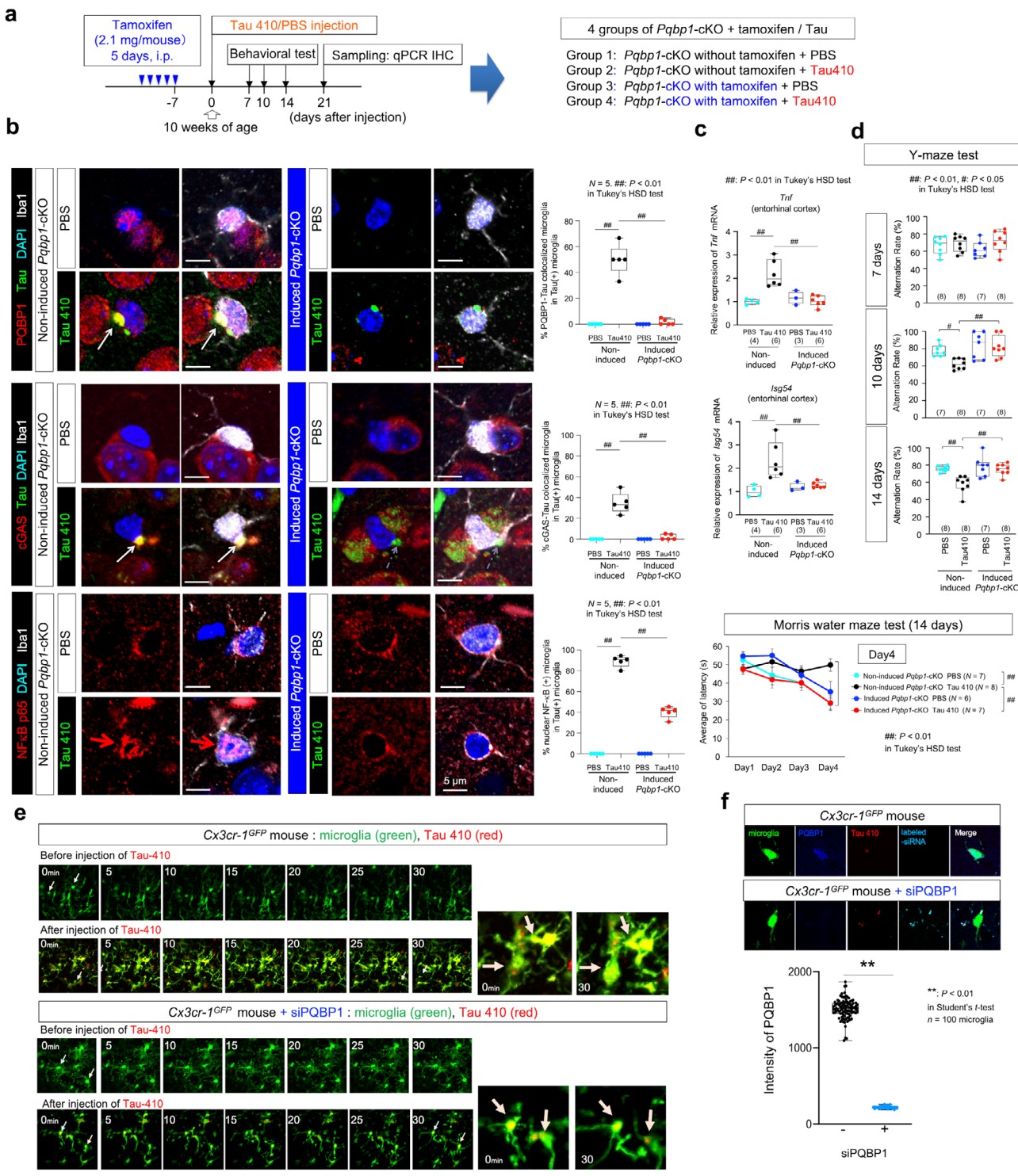

Collectively, our results indicate that tau activates microglia by tau monomer-induced PQBP1-dependent pathway (LRP1–PQBP1 pathway) and PQBP1-independent pathway (TREM2–ERK/PI3K pathway) that trigger neuronal death in parallel (Fig. 10b, left panels). Conversely, neuronal cell death directly induced by incorporation of tau polymer (especially of disease mutant tau) triggered microglia activation (Fig. 10b, right panels). The vicious circle generated by activated microglia and damaged neurons based on critical molecules, including LRP1, TREM2, and PQBP1, would definitely contribute to AD and tauopathy pathologies and would be critical therapeutic targets.

## Discussion

This study revealed that the cGAS–STING pathway works similarly both in virus infections and neurodegenerative diseases (Supplementary Fig. 10). Unexpectedly, intracellular dynamics of tau protein such as direct incorporation into rough ER was also homologous to that of corona viruses including SARS-CoV-2 of COVID-19 (refs. [46,47]) (Supplementary Fig. 10). Moreover, results obtained in this study have provided us with a further perspective that tau-propagation-induced pathologies of tauopathy are based not only on prionoid propagation among neurons, which possesses a kind of specificity in propagation pattern[38–40], but also

**Fig. 6 PQBP1 is essential for in vivo activation of microglia by extrinsic Tau. a** Protocol of Tau 410 monomer (2.2 μg/μl, 1 μl/injection, 4.4 μg/mouse) or equivalent PBS injection to bilateral entorhinal cortices of non-induced or tamoxifen-induced *Pqbp1*-cKO mice. Four groups were made for analyses. i.p. intra-peritoneal injection, qPCR quantitative RT-PCR, IHC immunohistochemistry. **b** Immunohistochemistry of Iba1-positive microglia with PQBP1-Tau colocalization, cGAS-tau colocalization, and NFκB nuclear translocation in injected areas of non-induced *Pqbp1*-cKO and tamoxifen-induced *Pqbp1*-cKO. (Right) Quantitative analyses of PQBP1-Tau colocalized microglia, cGAS-tau colocalized microglia, and activated microglia (nuclear NFκB-positive) among Tau-incorporating microglia. $N = 5$. *P* value: [Upper] $3.55e^{-9}$ (non-induced), $5.71e^{-9}$ (Tau410). [Middle] $1.00e^{-7}$ (non-induced), $1.00e^{-7}$ (Tau410). [Lower] $2.29e^{-14}$ (non-induced), $6.98e^{-12}$ (Tau410). **c** RT-qPCR of induction of *Tnf* and *Isg54* by Tau 410 in entorhinal cortex from non-induced or tamoxifen-induced *Pqbp1*-cKO mice. $N = (4, 6, 3, 6)$. *P* value: [Upper] 0.0011 (non-induced), 0.0005 (Tau410). [Lower] 0.0031 (non-induced), 0.0080 (Tau410). **d** Y-maze test at 7, 10, and 14 days and Morris water maze test at 14 days after tau injection into bilateral entorhinal. Data points (Morris water maze test): mean ± SEM. Y-maze: $N = (8, 8, 7, 8)$ for 7 days, $(7, 8, 7, 8)$ for 10 days and $(8, 8, 7, 8)$ for 14 days. Morris water maze: $N = (7, 8, 6, 7)$. *P* value: [Y-maze; 10 days] 0.0439 (non-induced), and 0.0055 (Tau410). [14 days] 0.0014 (non-induced), and 0.0018 (Tau410). [Morris water maze, Day4] 0.0043 (non-induced), and 0.0048 (Tau410). **e** In vivo live imaging of microglia by two-photon microscopy with B6.129P2(Cg)-*Cx3cr1*$^{tm1Litt}$/J (*Cx3cr-1*$^{GFP}$) mice in which microglia were labeled by the endogenous *Cx3cr1* enhancer/promoter-driven EGFP protein. **f** Immunohistochemistry confirmed PQBP1-KD by siRNA-PQBP1 injection. (Graph) Quantification of PQBP1 signal intensity/cell. Data points: mean ± SEM. $n = 100$ microglia. $P = 1.71e^{-102}$, **$P < 0.01$ in two-sided Student's *t*-test. Box plots show the median, quartiles, and whiskers that represent data outside the 25th to 75th percentile range. Tuple of *N* numbers indicates PBS/non-induced, Tau410/non-induced, PBS/induced *Pqbp1*-cKO, and Tau410/induced *Pqbp1*-cKO), respectively. For multiple group comparisons, Tukey's HSD test was performed (##$P < 0.01$, #$P < 0.05$).

on substance diffusion between neurons and microglia, which by itself does not define a specificity for expansion pattern in the brain.

PQBP1 was the firstly identified intracellular receptor for HIV in the cGAS–STING pathway[8], which was followed by the discovery of various intracellular receptors such as ZCCHC3 (ref. [9]) and G3BP1 (ref. [10]) for extrinsic DNAs. Meanwhile, since these proteins have also protein–protein interactions, if the intracellular receptors include neurodegenerative disease proteins such as PQBP1, the cGAS–STING pathway in microglia might act similarly to extrinsic viruses[74] and to extrinsic neurodegenerative disease proteins, as hypothesized previously[37]. Given that the hypothesis was proven by this study, the scheme could be further expanded to huntingtin, ataxin-1 and ataxin-3 via PQBP1 (refs. [12,13,41]) and possibly to other target neurodegenerative proteins via certain intracellular receptors in the cGAS–STING pathway that may be discovered in the future. Interestingly, human FUS, TDP43, hnRNPA1, and hnRNPA2B1, which cause amyotrophic lateral sclerosis (ALS)[75], contain multiple prolines (https://www.uniprot.org/), and mutations around which are related to the severity of disease[76].

The second issue raised by this study could also be considered as the relationship between intercellular transmission among neurons and one-way transmission of tau proteins to microglia. Tau and some other neurodegenerative disease proteins propagate in the brain, at least in cell and animal model experiments[38–40]. The mechanism of how extracellular tau proteins are taken up into neurons is not exactly known, while fundamental mechanism such as phagocytosis, pinocytosis, receptor-mediated uptake, exosome–ectosome, nanotube, and so on are suggested[77]. Endocytosis, including phagocytosis, pinocytosis, and receptor-mediated uptake[78], is fundamentally shared between microglia and neurons. Our experiments showed that the responses of microglia to different isoforms of tau such as 4R-tau (Tau 441) and 3R-tau (Tau 410) are similar, in contrast to inter-neuronal prionoid transmission patterns specific to tau species[49,79]. Intriguingly, the cGAS–STING system does not require aggregation of disease proteins, which is template-directed misfolding/aggregation and dependent on difference of isoforms/species[49,79].

Genetic studies have revealed that TREM2 is a risk factor of AD[80,81], and TREM2 mediates Aβ-sensing of microglia[82]. A comprehensive single cell RNA-seq study revealed a disease-specific group of microglia, which are activated by TREM2-dependent and TREM2-independent mechanisms in mouse AD model and human AD patient brains[51]. In addition, such a group

of disease-associated microglia takes up Aβ to intracellular/phagocytic particles[51]. Therefore, an important question would be whether the PQBP1-cGAS–STING pathway-dependent activation of microglia by non-aggregated tau revealed in this study corresponds to the TREM2-independent activation of microglia. Presumably but not exclusively LPL, a marker of the disease-specific microglia[51], was expressed in microglia whose activation was mediated by PQBP1 (Supplementary Fig. 8). TREM2 signaling also leads to NFκB activation[83] and our data revealed that deficiency of PQBP1-cGAS–STING pathway leads to 50% suppression of NFκB activation (Figs. 4 and 5), supporting the hypothesis that the PQBP1–cGAS–STING pathway induced by tau and TREM2-pathway induced by Aβ are complementary mechanisms involved in the activation of microglia and their conversion to disease-specific microglia in AD brains. Consistently, our genetic analysis of double KO of *Pqbp1* and *Lrp1* or *Trem2* in microglia by using Luciferase and RFP assays to evaluate their effects (Fig. 8b–d) indicated that LRP1–PQBP1 and TREM2-ERK/PIK3 pathways transduced two lines of cell signaling from tau to NFκB (Fig. 8e).

Our results collectively revealed the monomer-dominant but 3R/4R-equivalent tau effect on PQBP1-mediated microglia activation (Figs. 1–7), which indirectly induced neuronal cell death (Fig. 9 and Supplementary Fig. 9). In addition, we revealed polymer-dominant and 4R/disease-mutant dominant tau effects that directly induced neuronal cell death after their incorporation into neurons (Fig. 9 and Supplementary Fig. 9). Our analysis of their effects on cognitive functions (Fig. 10a), although it was performed under a restrictive condition regarding time window, spatial distribution of pathology, and tau dose dependency, suggested that PQBP1 largely contributes to the indirect pathway while less remarkably to the direct pathway. We previously reported that SRRM2 phosphorylation-induced PQBP1 deficiency in the nucleus causes synapse reduction and cognitive impairment[35]. In this regard, it remains possible that incorporated tau causes PQBP1 dysfunction though an abnormal interaction (Fig. 10b), and the question should be investigated further in the future.

The results of this study indicated that microglia-specific depletion of PQBP1 could be used to develop therapeutics against tau-mediated neurodegenerative diseases. Meanwhile, we have previously shown that impairment of PQBP1 in neurons leads to synaptic loss in the early stage of AD, and adeno-associated virus-PQBP1 (AAV-PQBP1) effects could be developed for therapeutics[35]. The current and previous findings could be coordinated since neuronal and microglial functions of PQBP1

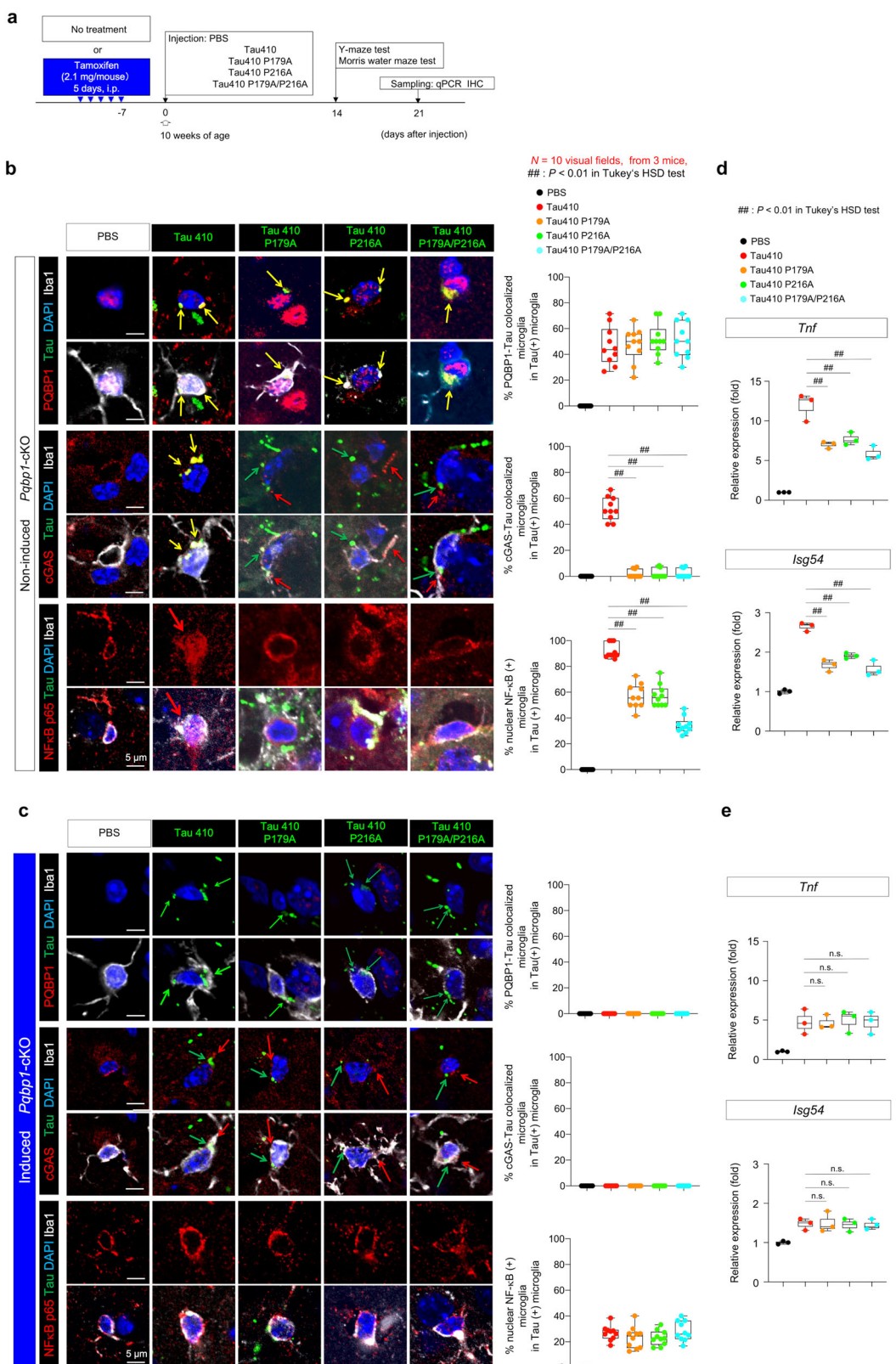

are naturally different. Since our western blot data indicated that AAV-PQBP1 upregulated PQBP1 protein in primary neurons but not in primary microglia (Supplementary Fig. 11), AAV-PQBP1 also remains as a seed therapy for AD[35] in addition to the microglia-specific KO or knockdown by utilizing anti-sense oligonucleotide, gene therapy, genome editing, and other newest techniques.

We observed tamoxifen-inducible microglia-specific homozygous *Pqbp1*-cKO mice for 6 months after administration of tamoxifen. These mice did not show any abnormality in their motor activity, eating, appearance, or behavior, though they were kept in the SPF condition. This might be because microglia activation is partially by the PQBP1-independent system (Fig. 8e). If PQBP1 deficiency in microglia is similarly safe for human

**Fig. 7 PQBP1-Tau interaction is essential for in vivo activation of microglia. a** Protocol of injection of Tau 410 or Tau 410 PQBP1-binding mutant monomers (0.5 μg/μl, 4.0 μl/injection, 4.0 μg/mouse) or equivalent PBS to bilateral entorhinal cortices of non-induced or tamoxifen-induced *Pqbp1*-cKO mice. Five groups were used for analyses. **b**, **c** Immunohistochemistry of Iba1-positive microglia with PQBP1 + tau, cGAS+tau, NFκB-p65+tau colocalization in injected area of non-induced *Pqbp1*-cKO, and tamoxifen-induced *Pqbp1*-cKO. Quantitative analyses of PQBP1-Tau colocalized microglia, cGAS-tau colocalized microglia, and activated microglia (nuclear NFκB-positive) among Tau-incorporating microglia are shown in the graphs on the right. *N* = 10 (visual fields, from 3 mice). ##*P* < 0.01 in Tukey's HSD test. **d**, **e** RT-qPCR of *Tnf* and *Isg54* induction by Tau 410 and the PQBP1-binding mutants with entorhinal cortex tissues prepared from non-induced or tamoxifen-induced *Pqbp1*-cKO mice. *N* = 3. ##*P* < 0.01, n.s.*P* > 0.05 in Tukey's HSD test (two-sided, multiple comparison). Box plots show the median, quartiles, and whiskers that represent data outside the 25th to 75th percentile range.

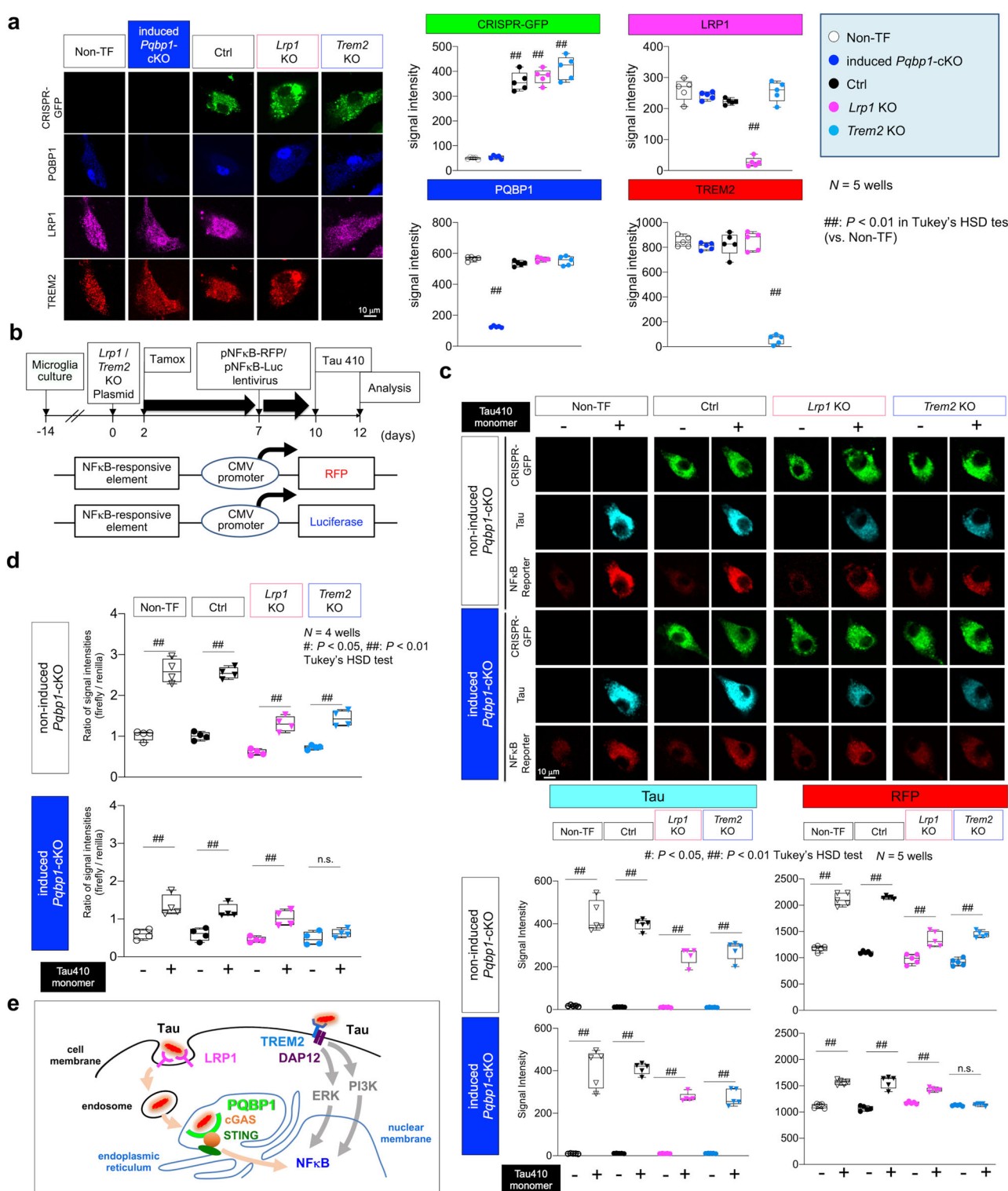

**Fig. 8 Genetic interaction of PQBP1 with LRP1 or TREM2 in microglia. a** Immunocytochemistry revealed high efficiency of tamoxifen-induced *Pqbp1*-cKO and D10A-mutant Cas9 nuclease/sgRNA-mediated KO of *Lrp1/Trem2*. Right graphs show quantitative analyses of signal intensities of target genes. Non-TF non-transfected. $N = 5$ wells. In the analysis of CRIPER-GFP signals: $P = 7.35e^{-12}$ (Ctrl vs Non-TF), $2.22e^{-12}$ (*Lrp1*-KO vs Non-TF), $3.31e^{-13}$ (*Trem2*-KO vs Non-TF); in the analysis of LRP1 signal: $P = 1.09e^{-11}$ (*Lrp1*-KO vs Non-TF); in the analysis of PQBP1 signals: $P = 1.92e^{-14}$ (*Pqbp1*-cKO vs Non-TF); in the analysis of TREM2 signals: $P = 6.76e^{-14}$ (*Trem2*-KO vs Non-TF); ##$P < 0.01$ in Tukey's HSD test (vs Non-TF). **b** Protocol for double KO of *Pqbp1-Lrp1* or *Pqbp1-Trem2* genes in primary microglia and their activation of NFκB by Tau 410. Tamox tamoxifen. **c** RFP signal assay to monitor activation of NFκB by adding Tau 410 in single KO microglia of *Lrp1* or *Trem2* or double KO microglia of *Pqbp1-Lrp1* or *Pqbp1-Trem2*. $N = 5$ wells. In tau signal assay of non-induced *Pqbp1*-cKO group: $P = 1.002e^{-13}$ [Non-TF (Tau+) vs (Tau−)], $1.005e^{-13}$ [Ctrl (Tau+) vs (Tau−)], $5.65e^{-11}$ [*Lrp1*-KO (Tau+) vs (Tau−)], $4.597e^{-12}$ [*Trem2*-KO (Tau+) vs (Tau−)]; in tau signal assay of induced *Pqbp1*-cKO group: $P = 1.005e^{-13}$ [Non-TF (Tau+) vs (Tau−)], $1.006e^{-13}$ [Ctrl (Tau+) vs (Tau−)], $1.73e^{-11}$ [*Lrp1*-KO (Tau+) vs (Tau−)], $1.98e^{-11}$ [*Trem2*-KO (Tau+) vs (Tau−)]; in RFP signal assay of non-induced *Pqbp1*-cKO group: $P = 4.60e^{-13}$ [Non-TF (Tau +) vs (Tau−)], $1.23e^{-13}$ [Ctrl (Tau+) vs (Tau−)], $4.93e^{-7}$ [*Lrp1*-KO (Tau+) vs (Tau−)], $1.15e^{-12}$ [*Trem2*-KO (Tau+) vs (Tau−)]; in RFP signal assay of induced *Pqbp1*-cKO group: $P = 1.005e^{-13}$ [Non-TF (Tau +) vs (Tau−)], $1.006e^{-13}$ [Ctrl (Tau+) vs (Tau−)], $1.73e^{-11}$ [*Lrp1*-KO (Tau+) vs (Tau−)], $0.9998$ [*Trem2*-KO (Tau+) vs (Tau−)]; #$P < 0.05$, ##$P < 0.01$ in Tukey's HSD test. **d** Luciferase assay to monitor activation of NFκB by adding Tau 410 to single KO microglia of *Lrp1* or *Trem2* or double KO microglia of *Pqbp1-Lrp1* or *Pqbp1-Trem2*. $N = 4$ wells. In non-induced *Pqbp1*-cKO group: $P = 2.20e^{-11}$ [Non-TF (Tau+) vs (Tau−)], $3.31e^{-11}$ [Ctrl (Tau+) vs (Tau−)], $7.40e^{-5}$ [*Lrp1*-KO (Tau+) vs (Tau−)], $5.76e^{-15}$ [*Trem2*-KO (Tau+) vs (Tau−)]; in induced *Pqbp1*-cKO group: $P = 8.13e^{-5}$ [Non-TF (Tau+) vs (Tau−)], $0.0013$ [Ctrl (Tau+) vs (Tau−)], $0.0028$ [*Lrp1*-KO (Tau+) vs (Tau−)], $0.9498$ [*Trem2*-KO (Tau+) vs (Tau−)]; #$P < 0.05$, ##$P < 0.01$ in Tukey's HSD test. **e** A hypothetical scheme of two pathways from extrinsic tau to activation of nuclear NFκB in microglia. Box plots show the median, quartiles, and whiskers that represent data outside the 25th to 75th percentile range.

patients, especially under the risk of infection, PQBP1 in microglia could be a target of therapeutics against AD, tauopathy, and other neurodegenerative diseases. Similar drug-inducible and microglia-specific cKO or conditional knockdown (cKD) technologies based on a viral vector could be developed.

## Methods

**Plasmid construction.** The DNAs encoding full-length PQBP1, PQBP1(1–94), Tau 441, and Tau(244–341) of Tau 410 were inserted into *Nde*I and *Bam*HI sites of a pOPTH plasmid, which was used to express the protein fused to an N-terminal His-tag in *E. coli*[20]. The DNAs encoding Tau 410, Tau 441, Aβ40, and α-synuclein were inserted into *Nde*I and *Bam*HI sites of a pOPHBL plasmid, which was used to express the protein fused to a sequence containing the N-terminal His-tag, a biotinylation-tag, and a lipoyl domain. The DNAs encoding Tau(1–150), Tau(151–197), Tau(198–243), and Tau(342–410) of Tau 410 were inserted into *Nde*I and *Bam*HI sites of a pOPHLT-s plasmid, which was used to express the proteins fused to a sequence containing an N-terminal His-tag, a lipoyl domain, and a TEV protease cleavage site[21].

Mutagenesis was carried out with pOPTH -Tau 410. The primer sets for P179A, P216A, P179A/P216A:
(P179A) forward: 5′-cccgcggctaaaaccccaccatcctct-3′,
reverse: 3′-ggttttagccgcgggaggcgttttggc-5′;
(P216A) forward: 5′-agcctggcaacaccaccgacccgtgaa-3′,
reverse: 3′-tggtgttgccaggctcggagtacgtga-5′.

The DNA encoding PQBP1(94–176) and PQBP1(193–265) were inserted into the *Bam*HI and *Sal*I sites of a pGEX6P-1 plasmid[18]. All constructs were verified by DNA sequencing.

A complete list of all primers used in this study has been added to Supplementary Information (Supplementary Table 3).

**Protein purification.** Rosetta(DE3)pLysS-competent cells (70956, Merck, Darmstadt, Germany) were transformed with pOPTH-Tau 410, pOPTH-Tau 410 P179A, pOPTH-Tau 410 P216A, pOPTH-Tau 410 P179A/P216A, pOPTH-Tau 441, pGEX-6P-1-PQBP1, pGEX-6P-1-PQBP1(1–94), pGEX-6P-1-PQBP1(94–176), pGEX-6P-1-PQBP1(193–265). After the *E. coli* culture reached an OD$_{600}$ of 0.5, IPTG was added at 1 mM and cells were incubated for another 2 h at 37 °C. The *E. coli* cells were collected, suspended in lysis buffer H (20 mM Tris-HCl pH 8.0, 100 mM NaCl, 20 mM Imidazole, protease inhibitor cocktail (539134, Calbiochem, San Diego, CA, USA)) for His-fusion proteins or in lysis buffer G (20 mM Tris-HCl pH 7.5, 100 mM NaCl, 1 mM DTT, protease inhibitor cocktail (539134, Calbiochem, San Diego, CA, USA)) for GST-fusion proteins, and sonicated for 8 min. His-fusion proteins were purified by Ni-NTA agarose (30230, Qiagen, Dusseldorf, Germany) and eluted with 8 ml of His elution buffer (20 mM Tris-HCl pH 8.0, 100 mM NaCl, 500 mM Imidazole, 1% glycerol, protease inhibitor cocktail (539134, Calbiochem, San Diego, CA, USA)), GST-fusion proteins were purified by glutathione-sepharose 4B (17075601, Cytiva, Marlborough, MA, USA)) and eluted with 2 ml of GST elution buffer (20 mM Glutathion, 50 mM Tris-HCl pH 8.0, 0.1% Triton X-100, protease inhibitor cocktail (539134, Calbiochem, San Diego, CA, USA)).

**Surface plasmon resonance analysis.** Tau proteins, Aβ40 and α-synuclein were fused to an N-terminal His-tag and a biotinylation-tag[20]. SPR measurements were performed with a BIAcore J instrument (GE Healthcare) at 25 °C[20]. The solution contained 10 mM Tris-HCl (pH 8.5), 150 mM NaCl, 3 mM EDTA, and 0.005% Tween-20.

Tau 410, Tau 441, Tau 410 P179A, Tau 410 P216A, Tau 410 P179A/P216A, or Tau 441 P301S were immobilized on CM5 sensor chips at a density of 1200 RU. Tau 441 P301S protein was purchased from Signal Chem (T08-56GN, Biotech, Richmond, BC, Canada). PQBP1 full-length, PQBP1(1–94), PQBP1(94–176), or PQBP1(193–265) were loaded at multiple doses for 120 s.

**NMR analysis.** Full-length tau and its fragments were expressed in *E. coli* BL21 (DE3) grown in M9 minimal medium supplemented with 15NH4Cl and 13C-glucose. 15N-labeled and 13C/15N-labeled proteins were purified with a Ni-NTA agarose resin (Qiagen, Valencia, CA) followed by gel filtration[84].

All tau proteins were purified by reverse-phase high-performance chromatography. The molecular weights of tau proteins were confirmed by MALDI-TOF mass analyses. The NMR samples contained 0.1–0.5 mM tau, 20 mM sodium phosphate (pH 7.0), 20 mM NaCl, 1 mM dithiothreitol, 1 mM NaN$_3$, 20 μM sodium 4,4-dimethyl-4-silapentane-1-sulfonate, and 10% D$_2$O. All NMR spectra were recorded on a Bruker Avance-800 NMR spectrometer with a cryoprobe at an acquisition temperature of 283 K.

**Preparation of tau oligomers and fibrils.** Tau 410 (T06-54N, Signal Chem, Biotech, Richmond, BC, Canada) and Tau 441 (T08-54N, Signal Chem, Biotech, Richmond, BC, Canada), Tau 410 P179A, Tau 410 P216A, Tau 410 P179A/P216A, and Tau 441 P301S (T08-56GN, Signal Chem, Biotech, Richmond, BC, Canada) dissolved in PBS at 2 μM containing 0.01 mg/ml heparin and 2 mM DTT were incubated at 37 °C with shaking. Incubation was continued for 5 or 10 days to generate tau oligomers or fibrils, respectively.

**Quantitative immunoprecipitation.** Primary microglia were cultured in the medium containing TAMRA-labeled Tau 410 (SP-502-100, R&D, Minneapolis, MN, USA) at a concentration of 25 nM for 3 days. Then, microglia were washed with ice-cold PBS, and $7 \times 10^5$ cells were harvested with 400 μl of lysis buffer (25 mM Tris-HCl, 150 mM NaCl, 1 mM EDTA, 1% NP40, 5% glycerol, and 0.5% protease inhibitor cocktail). Lysates were rotated for 60 min at 4 °C and then centrifuged at $366 \times g$ for 10 min. Supernatants were incubated with 1 μg of rabbit anti-PQBP1 (1:80, sc-32910, FL-265, Santa Cruz Biotechnology, Dallas, TX, USA), mouse anti-tau (1:400, MAB361, Merck, Darmstadt, Germany), or human IgG (1:400, 12000C, Thermo Fisher Scientific, Waltham, MA, USA) for 48 h at 4 °C, and incubated with Protein G Sepharose (GE Healthcare, Buckinghamshire, UK) for 1 h at 4 °C. Protein G Sepharose beads were precipitated, washed with lysis buffer three times, and the fluorescence intensity of TAMRA-labeled Tau 410 was measured on a FLUOstar OPTIMA-6 microplate reader (BMG Labtech, Durham, NC, USA) using 540 nm excitation and 590 nm emission wavelengths. For western blot, the samples were suspended in sample buffer (62.5 mM Tris-HCl, pH 6.8, 2% (w/v) sodium dodecyl sulfate (SDS), 2.5% (v/v) 2-mercaptoethanol, 10% (v/v) glycerol, and 0.0025% (w/v) bromophenol blue), left for 1 h at 37 °C, and boiled in sample buffer immediately before SDS-polyacrylamide gel electrophoresis (PAGE).

The list of antibodies used in this study with their validations on manufacturers' website has been added to Supplementary Information (Supplementary Table 4).

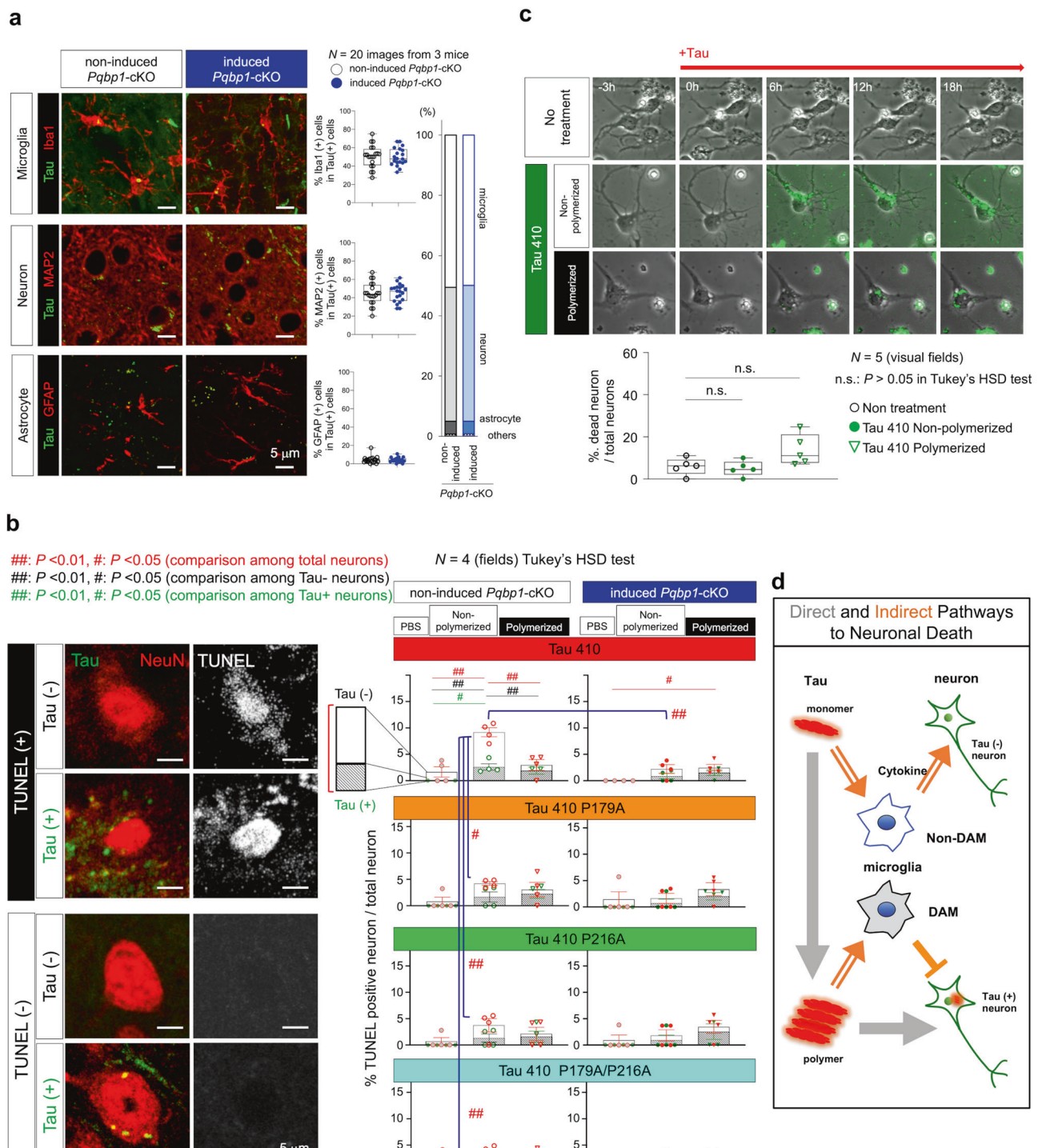

**Immunoprecipitation in R6/2 mice**. To examine binding of tau and PQBP1 in R6/2 mice, 10 mg of cerebral cortex was lysed by 200 μl of RIPA buffer (10 mM Tris-HCl pH 7.5, 150 mM NaCl, 1 mM EDTA pH 8.0, 1% Triton X-100, 0.1% SDS, and 0.1% deoxycholate). The lysate containing 300 μg protein was incubated with 1 μg of rabbit anti-PQBP1 antibody (1:200, A302-801A, Bethyl, Montgomery, TX, USA) or mouse anti-tau antibody (1:200, ab80579, Abcam, Cambridge, UK) for 2 h at 4 °C, and incubated with Protein G Sepharose (GE Healthcare, Buckinghamshire, UK) for 2 h at 4 °C. After centrifugation, the Protein G Sepharose pellet was washed three times with lysate buffer, boiled at 95 °C for 10 min, and were subjected to SDS-PAGE.

**Generation of *Pqbp1*-cKO mice**. To generate microglia-specific and tamoxifen-inducible conditional knockout mouse of *Pqbp1* (*Pqbp1*-cKO) on the C57BL/6 background, tamoxifen-inducible *Cx3cr1*^CreER/+ male mice were crossed with

*Pqbp1*^floxX/+ female mice. The double heterozygous mice were further crossed to obtain male: *Cx3cr1*^CreER/CreER/*Pqbp1*^floxX/Y, female: *Cx3cr1*^CreER/CreER/*Pqbp1*^floxX/floxX. *Cx3cr1*^CreER mice (B6.129P2(C)-*Cx3cr1*^tm2.1(cre/ERT2)Jung/J) were purchased from The Jackson Laboratory (Bar Harbor, ME, USA). *Pqbp1*-floxed mice were generated as described previously[34]. In all, 100 μg/g tamoxifen (080M1292V, Sigma-Aldrich, St Louis, MO, USA) dissolved in corn oil (C8267, Sigma-Aldrich, St Louis, MO, USA) was injected intraperitoneally into *Pqbp1*-cKO mice once a day for consecutive 5 days to induce depletion of *Pqbp1* gene in microglia in vivo.

**Animal housing conditions**. The mice were maintained at 22 °C, under suitable humidity (typically 50%), and with a 12-h dark/light cycle. Detailed information such as age and sex of the mice is listed in Supplementary Information (Supplementary Table 5).

**Fig. 9 Direct and indirect pathways mediating neuronal toxicity of Tau. a** Immunohistochemistry of Tau-injected mice revealed most of the non-microglia cells incorporating tau were MAP2-positive neurons but not GFAP-positive astrocytes. The ratios of Tau-incorporating cells were equivalent in non-induced and tamoxifen-induced *Pqbp1*-cKO mice, as the basis for analyses in (**b**). $N = 20$ images from three mice. Statistical test: Student's *t*-test. **b** Frequencies of neuronal cell death (TUNEL-positive and NeuN-positive dying neurons) by addition of normal or PQBP1-binding mutant Tau 410 were changed in Tau-negative neurons in addition to that of Tau-positive neurons. Indirect effect of Tau 410 monomer on Tau-negative neurons was suppressed by PQBP1-binding mutation or by tamoxifen-induced *Pqbp1*-cKO, while the direct effect of Tau 410 polymers on Tau-positive neurons was not affected by PQBP1-binding mutation or by tamoxifen-induced *Pqbp1*-cKO. TUNEL terminal deoxynucleotidyl transferase-mediated dUTP nick end labeling. Values in each group are presented as mean ± SEM. $N = 4$ fields. $^{\#}P < 0.05$, $^{\#\#}P < 0.01$ in Tukey's HSD test. **c** Direct effect of monomer and polymerized Tau 410 on primary neurons. Non-polymerized Tau 410 was not toxic to neurons in the absence of microglia. $N = 5$ visual fields. $^{n.s.}P > 0.05$ in Tukey's HSD test. **d** Direct and indirect pathways from extracellular tau to neuronal damage. Tau polymer directly damages neurons, while tau monomer is incorporated into LPL-negative non-DAM microglia (Supplementary Fig. 8), and indirectly damages neurons via cytokines. Though not included in this study, DAM around Aβ plaques might play a different role to prevent tau seeding. Box plots show the median, quartiles, and whiskers that represent data outside the 25th to 75th percentile range.

**Preparation and primary culture of microglia**. Mouse primary microglia were prepared from *Pqbp1*-cKO mice at P1–P3. Cortices were dissected, minced, and digested with 0.05% Trypsin-EDTA (#25200056, Thermo Fisher Scientific Inc., MA, USA) at 37 °C for 10 min. Trypsinization was stopped by adding 10% fetal bovine serum, and 10 mg/ml DNase was added to digest the sticky DNA. The cells were collected by centrifugation at $600 \times g$ for 3 min, dissociated by pipetting and passage through a 40 μm cell strainer (#22-363-548, Thermo Fisher Scientific Inc., MA, USA), and seeded into a T-75 culture flask in primary microglia culture medium (MGC57, Cosmo Bio Co., Ltd, Tokyo, Japan) supplemented with 1% penicillin/streptomycin. Medium was replaced once a week. At DIV 14-21, microglial cells were isolated from confluent mixed glial cultures by shaking (80 r.p.m. for 60 min at 37 °C) and centrifuged at $600 g$ for 10 min. The floating cells were collected, pelleted, and reseeded on 12-well plates. Microglial cells were used after another 24 h.

*Cx3cr1*$^{CreER}$ was induced by incubation for 5 days in medium with 4-hydroxytamoxifen (SML1666, Sigma-Aldrich, St Louis, MO, USA) diluted to a final concentration of 10 μg/ml.

Transfections of siRNAs were performed with Viromer BLUE (TT100300, OriGene, Rockville, MD, USA) according to the manufacturer's protocol. siRNAs against cGAS (MB21D1) (SASI_Mm01_00129826, Sigma-Aldrich, St Louis, MO, USA) and STING (TMEM173) (sc-154411, Santa Cruz Biotechnology, Dallas, TX, USA) were purchased.

Tau 410 (T06-54N, Signal Chem, Biotech, Richmond, BC, Canada), Tau 410 P179A, Tau 410 P216A, Tau 410 P179A/P216A, Tau 441 (T08-54N, Signal Chem, Biotech, Richmond, BC, Canada), and Tau 441 P301S (T08-56GN, Signal Chem, Biotech, Richmond, BC, Canada) were diluted in medium to a concentration of 25 nM and incubated for another 2 days.

**Immunocytochemistry**. Cultured cells were fixed with 2% paraformaldehyde (prepared in PB) at room temperature for 30 min. After fixation, cells were treated with 0.1% Triton X-100 in PBS for 10 min, blocked with PBS containing 1% bovine serum albumin and 0.1% Triton X-100 for 30 min at room temperature and incubated with primary antibodies diluted in the blocking buffer. The primary antibodies were anti-PQBP1 antibody (1:250, FL-265, Santa Cruz Biotechnology, Dallas, TX, USA), anti-NFκB p65 Antibody (C-20) (1:250, sc-372, Santa Cruz Biotechnology, Dallas, TX, USA), anti-phospho-tau antibody (1:1000, AT-8, Innogenetics, Ghent, Belgium), and anti-cGAS rabbit antibody (1:500, ABF124, Merck, Darmstadt, Germany).

**Luciferase assay**. Primary microglia from *Cx3cr1*$^{CreER/CreER}$/*Pqbp1*$^{floxX/Y}$ male mouse or female: *Cx3cr1*$^{CreER/CreER}$/*Pqbp1*$^{floxX/floxX}$ female mouse were incubated with 4-hydroxytamoxifen (10 μg/ml, SML1666 Sigma-Aldrich) for 5 days. Then the NFκB sensor primary microglia were developed by transduction with NFκB Reporter lentivirus ($1 \times 10^5$ TU) (CLS-013L-8, Qiagen) to $1 \times 10^4$ primary microglia. After 7 days of transduction, Tau 410 (T06-54N; Signal Chem, Biotech, Richmond, BC, Canada) and Tau 441 (T08-54N, Signal Chem, Biotech, Richmond, BC, Canada) were added to the medium for 48 h to a final concentration of 25 nM. Luciferase assays were performed using the Dual-Glo Luciferase assay system (Promega, WI, USA).

**Stereotaxic surgery and Tau injection**. Mice were deeply anesthetized with iso-flurane (Zoetis Japan, Tokyo, Japan) and immobilized in a stereotaxic frame. In Fig. 6, 1 μl of Tau 410 (2.2 μg/μl, 4.4 μg/bilateral injections, SP-502, Boston Biochem, Cambridge, MA, USA) was injected into bilateral EC at the following coordinates (from bregma): anterior/posterior (A/P): −4.7 mm, medial/lateral (M/L): ±3.3 mm, dorsal/ventral (D/V): −2.0 mm from brain surface. In Figs. 7, 9, 10 and Supplementary Fig. 9, non-polymerized and polymerized Tau 410, Tau 410 P179A, Tau 410 P216A, Tau 410 P179A/P216A, Tau 441, and Tau 441 P301S were labeled with Alexa Fluor 488 Microscale Protein Labeling Kit (A30006, Thermo Fisher Scientific, Waltham, MA, USA), and 4 μl of each labeled tau (0.5 μg/μl, 4 μg/bilateral

injections) were injected at a rate of 0.2 μl/min via a glass micropipette. To avoid any suction effect of the solution injected, micropipettes were left in place for an additional 5 min before being slowly withdrawed. Afterwards, the incision was closed by suture.

**Mouse behavioral analysis**. Spatial memory during exploratory behavior was assessed by the Y-maze test, using Y-shape maze with three identical arms inter-connected at an angle of 120° (YM-3002, O'HARA & Co., Ltd, Tokyo, Japan). After mice were habituated to the procedure room at least 30 min prior to testing, they were put at the end of one arm and allowed to explore freely through the maze during an 8 min session. The percentage of spontaneous alterations (indicated as an alteration score) was calculated based on the sequence of arm entries, dividing the number of entries into a new arm different from the previous one by the total number of transfers from one arm to another.

In the Morris water maze test, mice were trained to escape from water by swimming onto a hidden platform. Water temperature was maintained at $21 \pm 1$ °C. Mice performed four training trials per day (60 s) for 4 days, and the latency to reach the hidden platform was recorded.

**Immunohistochemistry**. Mouse brains injected with non-fluorescently labeled tau protein were fixed in 4% paraformaldehyde for 12 h and embedded in paraffin. Sagittal or coronal sections (thickness, 5 μm) were de-paraffinized in xylene, re-hydrated, dipped in 0.01 M citrate buffer (pH 6.0), and microwaved at 120 °C for 15 min. Mice injected with fluorescently labeled tau protein were perfused with 0.9% NaCl prior to brain tissue fixation, followed by tissue processing to form 25-μm-thick frozen sections. Sections were blocked with 10% FBS containing PBS. Immunohistochemistry was performed using the following primary antibodies: mouse anti-PQBP1 (1:200, sc-374260, Santa Cruz Biotechnology, Dallas, TX, USA); rabbit anti-PQBP1 (1:200, A302-801A, Bethyl Laboratory, Montgomery, TX, USA); rabbit anti-Iba1 (1:1000, 019-19741, Wako, Osaka, Japan); goat anti-Iba1 (1:500, 011-27991, Wako, Osaka, Japan); goat anti-Iba1 (1:500, ab107159, Abcam, Cambridge, UK); rabbit anti-NFκB p65 (C-20) (1:250, sc-372, Santa Cruz Biotechnology, Dallas, TX, USA); mouse anti-phospho-tau (1:1000, AT-8, Innogenetics, Ghent, Belgium); mouse anti-tau (1:500, MA5-15108, Thermo Fisher Scientific, Waltham, MA, USA); rabbit anti-cGAS (1:500, ABF124, Merck, Darmstadt, Germany); mouse anti-LPL (1:100, ab21356, Abcam, Cambridge, UK); rabbit anti-MAP2 (1:1000, ab32454, Abcam, Cambridge, UK); mouse anti-GFAP-Cy3 (1:5000, C9205, Sigma-Aldrich, St. Louis, MO, USA); mouse anti-NeuN (1:1000, ab104224, Abcam, Cambridge, UK). Secondary antibodies were as follows: donkey anti-goat IgG Alexa568 (1:1000, A11057, Molecular Probes, Eugene, OR, USA); donkey anti-mouse IgG Alexa488 (1:1000, A21202, Molecular Probes, Eugene, OR, USA); donkey anti-rabbit IgG Alexa647 (1:1000, A31573, Molecular Probes, Eugene, OR, USA). Nuclei were stained with DAPI (0.2 μg/ml in PBS, D523, DOJINDO Laboratories, Kumamoto, Japan).

For nick end labeling, frozen sections were washed three times with PBS containing 0.1% Tween-20 (PBST) at room temperature. The sections were incubated with labeling reaction mix [Biotin-16-dUTP (11093070910, Roche, Mannheim, Germany) and terminal transferase (03333574001, Roche, Mannheim, Germany)] at 37 °C for 2 h, washed with PBST three times, and then incubated with Alexa Fluor 633-conjugated streptavidin (S21375, Thermo Fisher Scientific) at room temperature for 1 h.

**Western blot analysis**. Mouse primary microglia in culture dishes were washed three times with PBS and dissolved in sample buffer containing 62.5 mM Tris-HCl, pH 6.8, 2% (w/v) SDS, 2.5% (v/v) 2-mercaptoethanol, 10% (v/v) glycerol, and 0.0025% (w/v) bromophenol blue. Samples were separated by SDS-PAGE, trans-ferred to Immobilon-P polyvinylidene difluoride membranes (Millipore, Burling-ton, MA, USA) using the semi-dry method, blocked with 5% milk in TBST (10 mM Tris-HCl pH 8.0, 150 mM NaCl, and 0.05% Tween-20), and reacted with the following primary and secondary antibodies diluted in Can Get Signal solution

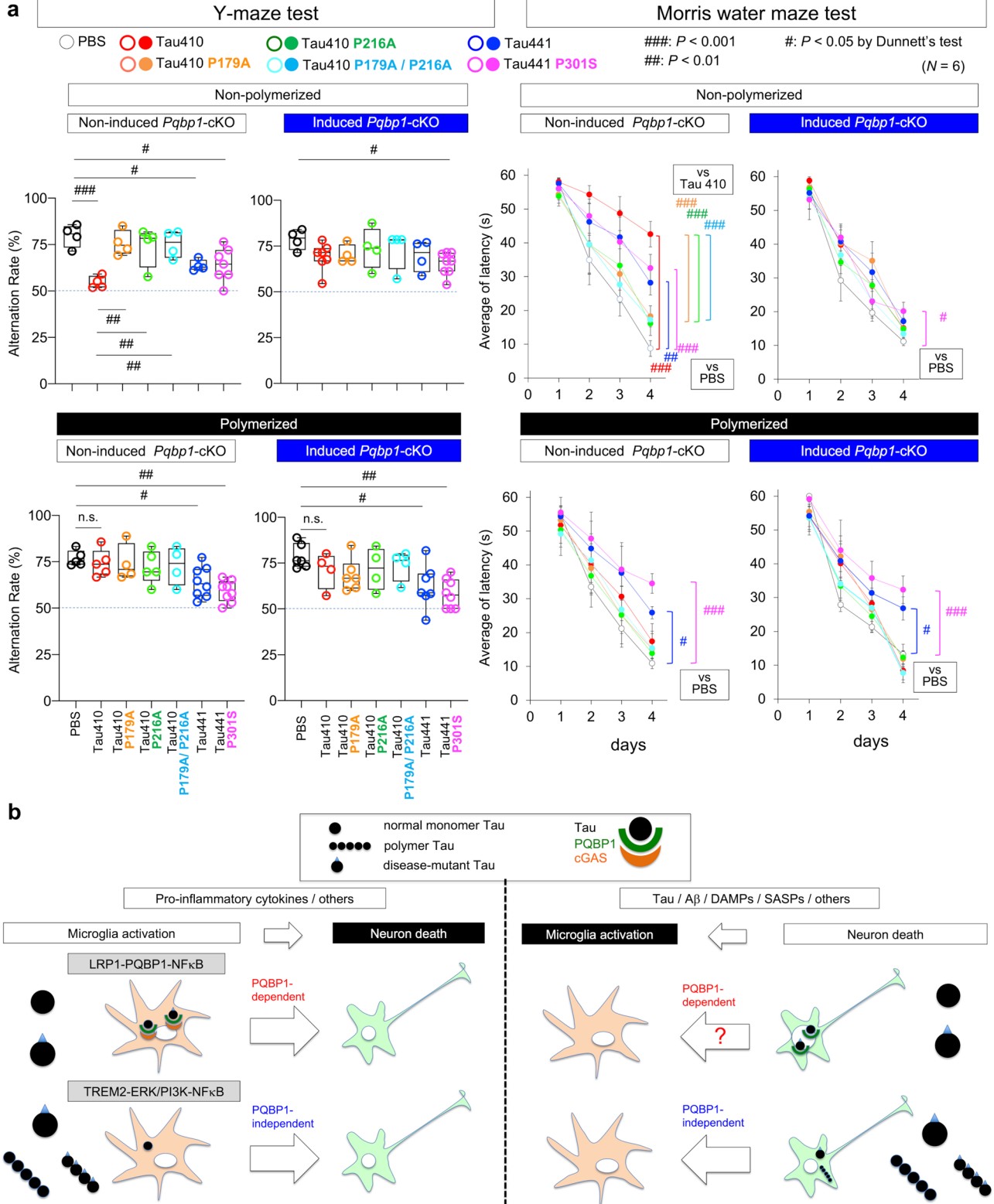

**Fig. 10 PQBP1 depletion in microglia recovers memory functions impaired by Tau. a** Memory dysfunction after injection of various species of tau in non-polymerized or polymerized state was evaluated by Y-maze test and Morris water maze test. Protocol from injection to memory function tests in various conditions is shown in Fig. 6a. Y-maze test: *N* number in each group is indicated in Source data. In Morris water maze test, values in each group are presented as mean ± SEM; *N* = 6 mice/group. #*P* < 0.05, ##*P* < 0.01, ###*P* < 0.001 in Dunnett's test. Box plots show the median, quartiles. and whiskers that represent data outside the 25th to 75th percentile range. **b** PQBP1-depndent and PQBP1-independent pathways mediating microglia activation to neuronal cell death or neuronal cell death to microglia activation. Different types of tau species trigger distinct pathways.

(Toyobo, Osaka, Japan as follows: mouse anti-tau (1:3000, MAB361, Millipore, Burlington, MA, USA); rabbit anti-PQBP1 (1:500, sc-32910, Santa Cruz Biotechnology, Dallas, TX, USA); rabbit anti-cGAS (1:1000, ABF124, Merck, Darmstadt, Germany); rabbit anti-STING (1:3000, 13647S, Cell Signaling Technology, Danvers, MA, USA); rabbit anti-phospho-Ser536-NFκB (1:1000, 3033S, Cell Signaling Technology, Danvers, MA, USA); rabbit anti-phospho-Ser396-IRF3 (1:1000, 4947S, Cell Signaling Technology, Danvers, MA, USA); mouse anti-GAPDH (1:5000, MAB374, Millipore, Burlington, MA, USA); mouse anti-tau (1:10,000, ab80579, Abcam, Cambridge, UK); rabbit anti-PQBP1 (1:1000, A302-801A, Bethyl, Montgomery, TX, USA); HRP-conjugated anti-mouse IgG (1:3000, NA931VA, GE Healthcare, Chicago, IL, USA); and HRP-conjugated anti-rabbit IgG (1:3000, NA934VS, GE Healthcare, Chicago, IL, USA). ECL Select Western Blotting Detection Reagent (RPN2235, Cytiva, Marlborough, MA, USA) and an Image-Quant LAS 500 luminescent image analyzer were used to detect proteins. Full-scan images are displayed in Source data file.

**Quantitative RT-PCR.** Total RNA was isolated from primary microglia at 2 days after heparin and Tau 410/441 (polymerized and not polymerized) incubation with NucleoSpin RNA XS (U0902A, TaKaRa, Shiga, Japan). Total RNA was isolated from entorhinal cortex tissues of 14 groups of *Pqbp1*-cKO mice at 21 days after tau proteins/PBS injection with RNeasy mini kit (74106, Qiagen, Limburg, The Netherlands). To eliminate genomic DNA contamination, on-column digestion of DNA was carried out with DNase I. Reverse transcription was performed using the SuperScript VILO cDNA Synthesis kit (11754-250, Invitrogen, Carlsbad, CA, USA). Quantitative PCR analyses were performed with the 7500 Real-Time PCR System (Applied Biosystems, Foster City, CA, USA) using the Thunderbird SYBR Green (QPS-201, TOYOBO, Osaka, Japan) and assessed by the standard curve method. The primer sequences were:

mouse *Tnf*, forward primer: 5′-TGCTTGTTGACAGCGGTCC-3′ and reverse primer: 5′-ACTGGCCATCGTGGAGGTAC-3′
mouse *Isg54*, forward primer: 5′-AGCAAGATGCACCAAGATGA-3′ and reverse primer: 5′-CTGTGTCAAAGCGCTCAAAG-3′
mouse *Ifnβ*, forward primer: 5′-GCCTTTGCCATCCAAGAGATGC-3′ and reverse primer: 5′-ACACTGTCTGCTGGTGGAGTTC-3′
mouse *Cxcl10*, forward primer: 5′-ATCATCCCTGCGAGCCTATCCT-3′ and reverse primer: 5′-GACCTTTTTTGGCTAAACGCTTTC-3′
mouse *Gapdh*, forward primer: 5′-TGAACGGGAAGCTCACTGG-3′ and reverse primer: 5′-TCCACCACCCTGTTGCTGTA-3′

The PCR conditions for amplification were 95 °C for 10 min for enzyme activation, 95 °C for 15 s for denaturation, and 60 °C for 1 min for extension (40 cycles). The expression levels of individual genes were normalized against the expression of GAPDH and calculated as a relative expression level.

**Live imaging of microglia in primary culture.** Mouse primary microglia were prepared from *Pqbp1*-cKO or C57BL/6 mice at P3. At DIV 14–21, microglial cells were isolated from confluent mixed glial cultures and reseeded on eight-well chambers. After 24 h, pEGFP-N1-PQBP1 was transfected into microglia using Viromer RED (TT100302, OriGene, Rockville, MD, USA) for 24 h. Time-lapse image was acquired with an Olympus FV10i-W laser-scanning microscope at ×60 magnification (Olympus, Tokyo, Japan) every 15 min for the next 42 h. TAMRA-labeled Tau proteins were added to the medium at 18 h post-transfection. The chamber was kept at 37 °C with 5% $CO_2$.

**Live imaging of microglia in in vivo mouse brains.** Before siRNA injection, siRNA was labeled with Label IT siRNA Tracker Cy5 Kit without Transfection Reagent (MIR7213, Mirus, WI, USA) according to the manufacturer's procedures. Under anesthesia with 1% isoflurane, 300 ng Cy5-labeled-siRNA (mouse PQBP1-siRNA, sc-38200, Santa Cruz Biotechnology, Dallas, TX, USA) in 1 μl volume was injected into the retrosplenial cortex (anteroposterior, −3.0 mm form bregma; lateral, 0.6 mm; depth, 0.5 mm) of *Cx3cr1-GFP* mouse (005582, Jackson Laboratory, Bar Harbor, MW, USA) at 21 weeks of age using in vivo jetPEI (201-10G, Polyplus-transfection, Illkirch, France). After 16 h, the skull was thinned with a high-speed micro-drill in the mouse splenial cortex. Then, the head of each mouse was immobilized by a head plate on a custom machine stage mounted on the microscope table. Two-photon imaging was performed at 5 min intervals for 30 min using a laser-scanning microscope system FV1000MPE2 (Olympus, Tokyo, Japan) equipped with an upright microscope (BX61WI, Olympus, Japan), a water-immersion objective lens (XLPlanN25xW; numerical aperture, 1.05), and a pulsed laser (MaiTaiHP DeepSee, Spectra Physics, Santa Clara, CA, USA). Then, after injecting 2.2 μg TAMRA-labeled Tau 410 (anteroposterior, −1.0 mm form bregma; lateral, 0.6 mm; depth, 0.5 mm), two-photon imaging was performed again at 5 min intervals for 30 min.

**Live imaging of neuron in primary culture.** Mouse primary cortical neurons were prepared from E17 C57BL/6J mouse embryos. Cerebral cortices were dissected and incubated with 0.05% trypsin in PBS at 37 °C for 15 min and dissociated by pipetting. The cells were passed through a 70 μm cell strainer (22-363-548, Thermo Fisher Scientific, MA, USA), collected by centrifugation, and cultured in neurobasal

medium (21103049, Thermo Fisher Scientific, Waltham, MA, USA) containing 2% B27 (17504044, Thermo Fisher Scientific, Waltham, MA, USA), 0.5 mM L-glutamine, and 1% penicillin/streptomycin (15140-122, Thermo Fisher Scientific, Waltham, MA, USA). Five days later, time-lapse images of neurons were acquired at ×60 magnification on an Olympus FV10i-W laser-scanning microscope (Olympus, Tokyo, Japan) at 1 h intervals for 24 h. The chamber was kept at 37 °C with 5% $CO_2$. The ratio of cell death patterns was counted 18 h after addition of labeled tau.

**Knockout of LRP1 and TREM2 in primary microglia.** In total, 0.6 μg of Control Double Nickase Plasmid (sc-437281, Santa Cruz Biotechnology, Dallas, TX, USA) LRP1 Double Nickase Plasmid (sc-421464-NIC, Santa Cruz Biotechnology, Dallas, TX, USA) and TREM2 Double Nickase Plasmid (sc-429903-NIC, Santa Cruz Biotechnology, Dallas, TX, USA) were electroporated into suspended microglia using 4D-Nucleofector (pulse program: CV-110) (4D-Nucleofector Core Unit, #AAF-1002B, LONZA, NJ, USA). The electroporated cells were cultured on Lab-Tek II chambered coverglass.

To test knockout efficiency, cells were fixed in 0.4% FA, blocked with 10% FBS in PBS for 60 min at RT, incubated with primary antibody for 60 min, and with secondary antibodies for 60 min at RT. The antibodies used for immunocytochemistry were diluted as follows: mouse anti-TREM2 (1:100, sc-373828, Santa Cruz Biotechnology, Dallas, TX, USA); rabbit anti-PQBP1 (1:150, Bethyl Laboratories, A302-801A, Montgomery,TX, USA); Cy5-conjugated anti-mouse IgG (1:500, 715-175-151, Jackson Laboratory, Bar Harbor, ME, USA); and Alexa Fluor 405–conjugated anti-rabbit IgG (1:1000, A48258, Molecular Probes, Eugene, OR, USA). For multiple co-staining, mouse anti-LRP1 antibody (1:250, sc-57353, Santa Cruz Biotechnology, Dallas, TX, USA) was labeled by Zenon Secondary Detection-Based Antibody Labeling Kits (Zenon Alexa Fluor 555 Rabbit IgG Labeling Kit, Z-25305, Thermo Fisher Scientific, Waltham, MA, USA).

To monitor NFκB activity, NFκB-RFP lentivirus (LVP966-P, GenTarget Inc., San Diego, CA, USA) driven by a minimal CMV promoter and NFκB transcriptional response element (NFκB-TRE) was infected to cells (1 × 10⁴ cells/well) at MOI of 5. At 72 h after infection, Tau 410 (T06-54N, Signal Chem, Biotech, Richmond, BC, Canada) labeled with Alexa Fluor 555 Microscale Protein Labeling Kit (A30007, Thermo Fisher Scientific, Waltham, MA, USA) was added to microglia at 25 nM and incubated for another 2 days. The fluorescent signal was analyzed by confocal microscopy (FV1200IXGP44, Olympus, Tokyo, Japan). The luciferase assay was also used to monitor NFκB activity similarly to the abovementioned method.

**Infection of AAV-PQBP1.** Mouse primary cortical neurons were prepared from E17 C57BL/6J mouse embryos. Cerebral cortices were dissected and incubated with 0.05% trypsin in PBS at 37 °C for 15 min and dissociated by pipetting. The cells were passed through a 70-μm cell strainer (22-363-548, Thermo Fisher Scientific, MA, USA), collected by centrifugation, and cultured in neurobasal medium (21103049, Thermo Fisher Scientific, Waltham, MA, USA) containing 2% B27 (17504044, Thermo Fisher Scientific, Waltham, MA, USA), 0.5 mM L-glutamine, and 1% penicillin/streptomycin (15140-122, Thermo Fisher Scientific, Waltham, MA, USA). Forty-eight hours later, the medium was changed to that containing 0.5 μM AraC (C3631, Sigma-Aldrich, St. Louis, MO, USA). Mouse primary microglia were prepared from C57BL/6 mice at P3 as described above. Primary microglia and primary neuron were infected by AAV1-CMV-hPQBP1 (ref. [35]) (MOI: 5000) in the culture medium for 5 days. For western blot, after culture medium was removed and the cells were washed with PBS, cells were added to lysis buffer (10 mM Tris-HCl, pH 7.5, 10 mM NaCl, 0.5 mM EDTA, 1% NP40, 0.5% protease inhibitor cocktail (539134, Calbiochem, San Diego, CA, USA)). Lysates were rotated for 60 min at 4 °C, and then centrifuged at 12,000 × g for 20 min at 4 °C. After adding 1% β-mercaptoethanol to the supernatants, the supernatants were rotated for 5 min at 4 °C, an equal volume of sample buffer (50 mM Tris-HCl, pH 6.5, 2% SDS, 10% glycerol, 0.005% BPB) was added, and the samples were boiled at 100 °C for 3 min before SDS-PAGE.

**Chemotaxis assay.** Chemotaxis assay of mouse primary microglia with and without PQBP1 depletion was performed with Boyden chambers (CBA-105, CytoSelectTM 96-Well Cell Migration Assay, Cell Biolabs, San Diego, CA, USA) following the manufacturer's protocol[85]. Overnight starvation was performed prior to running the assay. Cell suspensions of microglia incubated with and without tamoxifen were seeded into the upper membrane chambers at 2.5 × 10⁴ cells/well in 100 μl serum free media. Media (150 μl) containing 10% FBS was added to the lower chambers. Then, cells were allowed to migrate for 4 h at 37 °C in an incubator enriched with 5% $CO_2$. At the end of the assay, migrated cells, which clung to the bottom side of the membrane, were dissociated by adding cell detachment buffer. Then, cells were lysed and quantified using CyQuant GR Fluorescent Dye by a FLUOstar OPTIMA-6 microplate reader (BMG Labtech, Durham, NC, USA) using 480 nm excitation and 520 nm emission wavelengths.

**Phagocytosis assay.** Phagocytic activity of mouse primary microglia with and without PQBP1 depletion was performed with the Phagocytosis Assay Kit (500290, Cayman Chemical, Ann Arbor, MI, USA) according to the manufacturer's

protocol[86]. Cells were seeded into a 96-well plate at $1 \times 10^4$ cells/well. After the induction of PQBP1 depletion as previously described, latex beads coated with rabbit IgG-FITC conjugates were diluted by culture medium at a final concentration of 1:100. Then cells were incubated with 200 μl media containing latex beads for 4 h followed by a 1-min incubation with trypan blue to quench non-phagocytosed bead fluorescence. The fluorescence intensity was measured by a FLUOstar OPTIMA-6 microplate reader (BMG Labtech, Durham, NC, USA) at 485 nm excitation and 535 nm emission wavelengths. Cells were then fixed and counterstained with anti-Iba1 and -PQBP1 antibodies.

**Statistics**. Statistical analyses for biological experiments were performed using Graphpad Prism 8, R version 3.6.2., or Microsoft Excel for Microsoft 365.

Biological data following a normal distribution are presented as the mean ± SEM. Tukey's HSD test was applied. The significance level was set at 1 or 5%.

A box plot was used to depict distribution of observed data, and the data are also plotted as dots. Box plots show medians, quartiles, and whiskers, which represent data outside the 25th–75th percentile range.

For Fig. 9b, a three-way ANOVA was used to examine mutual effects among the examination conditions (tau mutation, polymerization, *Pqbp1*-cKO) on neuronal cell death. No interaction was detected between tau mutation and polymerization ($P = 0.1336$), while interaction was detected between *Pqbp1*-cKO and polymerization ($P = 0.0022$) or between *Pqbp1*-cKO and mutation ($P = 0.0408$). Therefore, a post hoc Tukey's HSD test was used to compare the number of dead cells in four types of non-polymerized PQBP1-binding mutations of tau. Significant difference was observed between Tau 410 and Tau 410 P179A ($P = 0.0144$), Tau 410 P216A ($P = 0.0028$), or Tau 410 P179A/P216A ($P = 0.0078$). For each type of tau, a post hoc Tukey's HSD test was used to compare the cell death numbers among all cases irrespective of polymerized/non-polymerized or induced/non-induced *Pqbp1*-cKO conditions. A significant difference was observed in Tau 410 (non-polymerized/non-induced vs non-polymerized/induced, $P = 0.0001$), but no significance was found in Tau 410 P179A, Tau 410 P216A, or Tau 410 P179A/P216A.

**Ethics for animal experiments**. All animal experiments were strictly performed according to the ARRIVE guidelines (Animal Research: Reporting in vivo Experiments) for the Care and Use of Laboratory Animals of the National Institutes of Health. The experiments were approved by the Committees on Gene Recombination Experiments and Animal Experiments of Tokyo Medical and Dental University (G2018-082C and A2019-218C2).

**Ethics for human experiments**. All experiments using human samples were conducted with informed consent and in accordance with the approved guidelines for human experimental research. All the experiments were approved by the Committee on Human Ethics of the Tokyo Medical and Dental University (O2014-005-13/O2020-002).

**Reporting summary**. Further information on research design is available in the Nature Research Reporting Summary linked to this article.

## Data availability
All relevant data are available within the article and its Supplementary Information files. UniProt database of subcellular locations was used for STING (https://www.uniprot.org/uniprot/Q86WV6) and for cGAS (https://www.uniprot.org/uniprot/Q8N884). Human FUS, TDP43, hnRNPA1, and hnRNPA2B1 protein sequences were retrieved from UniProt (https://www.uniprot.org/). Source data are provided with this paper.

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

## Acknowledgements
We thank Huang Yong, Mao Takata, and Sumire Takayama for technical assistance. This work was supported by Grant-in-Aid for Scientific Research from Japan Society for Promotion of Science (JSPS) (16H02655; 19H01042) and a Grant-in-Aid for Scientific Research on Innovative Areas (Foundation of Synapse and Neurocircuit Pathology, 22110001/22110002) from the Ministry of Education, Culture, Sports, Science and Technology of Japan (MEXT) to H.O.

## Author contributions
M.J., H.S., H.T. and K.F.: data curation and writing—original draft and revised draft. H.H.: computational analysis and writing—original draft and revised draft. T.O., S.O. and K.N.: data curation. Y.Y. X.J., and K.K.: data curation. M.M.: resources, data curation and writing—original draft. H.O.: conceptualization, supervision, funding acquisition, project administration, and writing original draft and revised draft.

## Competing interests
The authors declare no competing interests.
