## [Peer Review File · Nature Communications]

Tau activates microglia via PQBP1-cGAS-STING pathway in the similar way as HIV activates dendritic cellsREVIEWER COMMENTS

Reviewer #1 (Remarks to the Author):

In this manuscript Jin et al. present a set of experiments showing that PQBP1, a previously identified cGAS partner, target cGAS to Tau protein to induce activation of cGAS-STING-IRF3 pathway leading to production of type1 interferon. The study is of importance since it shows that cGAS, through PQBP1, can also sense misfolded protein to induce inflammatory response. The experiments are well designed and the results support the conclusions. The in vivo data strengthen and validate the ex vivo experiments. This reviewer is highly supportive for the publication of the manuscript.

Figure 1d: a representative immunoblot analyses should be shown.

ImmunoFluorescent panels should be accompanied by statistical graph.

Reviewer #2 (Remarks to the Author):

Jin et al. propose a novel mechanism to explain tauopathy and microglial activation in Huntington's disease (HD) and spinocerebellar ataxia type 1 (SCA1). First, they determined biophysical interaction between PQBP1 and different Tau fragments by SPR and NMR. Second, they showed cellular interactions between PQBP1 and phagocytosed tau fragments by co-IP and fluorescence imaging. Moreover, they assessed how Tau feeding and PQBP1 knockout/knockdown impact primary microglial activation through NF- κ B nuclei translocation, luciferase reporter, TNF α mRNA, etc. Finally, they showed in vivo data whereas conditional PQBP1 KO in microglia seems to impair microglia activation without interfering with Tau uptake by other cells. This is the first report that attempts to link PQBP1-cGAS-STING signaling with Tau binding directly. While the overall story is novel, I have some concerns and advice for additional experiments that may consolidate these findings.

Major concerns:

1. The authors proposed that PQBP1 directly recognizes extrinsic Tau phagocytosed by microglia and increases microglial activation through the cGAS-STING pathway. However, I am not sure how much does PQBP1-Tau interaction contribute to microglial activation beyond PQBP1 interactions with other polyQ proteins. Data in this manuscript showed that Tau 4R bind less to PQBP1 compared with 3R and induced less TNF α , which is counterintuitive as the tauopathy in HD pathology is associated with 4R repeats. Do the authors have an explanation on this? In other words, the authors show that PQBP1 can bind Tau, but this does not demonstrate that PQBP1 does bind in vivo.

2. The in vivo model is based on a Tau injection model where overall microglial activation was not impressive (<8% according to figure 5b, middle panels). The authors may utilize a disease mouse model to improve the results of this interaction in vivo. A direct evidence to show the PQBP1-Tau interaction in vivo occurs in a Huntington's disease model such as R2/6 mice might help.

3. Recently, LRP1 expressed by microglia has been demonstrated as a dominant Tau receptor. Since results in this manuscript suggest that PQBP1-cGAS-STING contributes to microglial activation after

Tau uptake. I would recommend the authors to include LRP1-KO in this study to further emphasize the sequential intervention of these pathways in Tau-uptake and microglia activation.

4. With current data, the direct physical interaction between PQBP1 and Tau is not conclusively demonstrated. The authors could have utilized their NMR data to introduce some mutations into Tau fragments to block the interaction between PQBP1 and Tau. Then, they could add this experimental group as negative control to show the change in co-localization and microglial response in vitro and in vivo.

5. Could the authors show the K_d values determined by their SPR assay? In figure 1B, the concentration of the proteins used seems too high.

6. In figure 2, could the authors also show quantitative analysis as what they did for figure 3c?

7. While the water maze results in figure 5d demonstrated that conditional PQBP1-KO in microglia rescues cognition defects caused by Tau injection, it was also clear that such KO in microglia causes massive Tau uptake by non-microglia cells (figure 5b, bottom panels). Given that PQBP1 is not exclusively expressed in microglia in the CNS, did the authors evaluate the identity and fate of

those cells that uptake Tau in a large amount?

8. The assessment of downstream of PQBP1-cGAS-STING pathway in this study could be extended to type I interferon response such as IFN β , CXCL10, etc., not just NF-kB nuclear translocation.

Reviewer #3 (Remarks to the Author):

The manuscript from Jin et al. examines the role of PQBP1 in Tau-mediated activation of microglia. The work begins with an evaluation of the Tau – PQBP1 interaction in microglia, and experiments are performed in primary mouse microglia, which is a physiologically appropriate system, and analytical methods and statistics are sound. The authors present data suggesting that the PQBP1 interaction with Tau occurs in the ER, which is where STING is also found, and show that the STING pathway is activated upon PQBP1 – Tau interaction in microglia. After documenting the necessity of PQBP1 for robust cGAS-STING activation in vitro, the study transitions to provocative in vivo experiments taking advantage of the authors' existing PQBP1 conditional knock-out mouse model. Using a co-injection paradigm of Tau monomer + tamoxifen to turn off PQBP-1 gene expression, they report impressive findings for read-outs including microglia activation, NF-kb nuclear translocation, cGAS-STING activation, and importantly, cognitive and memory function.

While it seems unlikely PQBP1 is expressed in microglia solely to be a sensor for misfolded Tau or other CNS insults, my assessment is that this work is novel and could represent a substantive advance.

Issues with this work that the authors need to address are as follows:

1) The authors note that PQBP1 depletion reduces NF-kb nuclear translocation by 50%, suggesting that other pathways are also operational. Can the authors elaborate upon what other pathways might be relevant here? They imply that in AD, there is another pathway stemming from amyloid-beta sensing involving TREM2, but can the authors test if TREM2 modulation alters the response to Tau in PQBP1 depleted microglia?

2) I find it curious that the microglia response to 3R-Tau and 4R-Tau is comparable in your experiments. Did you also consider evaluating Tau carrying FTD-linked mutations or other relevant functional variants?

3) One overriding question is the normal function of PQBP1 in microglia. Have you carefully analyzed other functions of microglia depleted of PQBP1? Are chemotaxis and phagocytosis intact?

4) Do mice depleted of PQBP1 in microglia develop any neurological phenotypes? In other words, should we be thinking of PQBP1 as a target for dosage reduction in Alzheimer's disease? I would like the authors to discuss whether or not PQBP1 is a viable therapeutic target in the final section of the paper.

REVIEWER COMMENTS

Reviewer #1 (Remarks to the Author):

In this manuscript Jin et al. present a set of experiments showing that PQBP1, a previously identified cGAS partner, target cGAS to Tau protein to induce activation of cGAS-STING-IRF3 pathway leading to production of type1 interferon. The study is of importance since it shows that cGAS, through PQBP1, can also sense misfolded protein to induce inflammatory response. The experiments are well designed and the results support the conclusions. The in vivo data strengthen and validate the ex vivo experiments. This reviewer is highly supportive for the publication of the manuscript.

>>> Thank you very much for kind evaluation of our manuscript. We really appreciate your kind efforts for the review of our manuscript.

Figure 1d: a representative immunoblot analyses should be shown. ImmunoFluorescent panels should be accompanied by statistical graph.

>>> Following the advice we show a representative blot in Figure 1e.

Reviewer #2 (Remarks to the Author):

Jin et al. propose a novel mechanism to explain tauopathy and microglial activation in Huntington's disease (HD) and spinocerebellar ataxia type 1 (SCA1). First, they determined biophysical interaction between PQBP1 and different Tau fragments by SPR and NMR. Second, they showed cellular interactions between PQBP1 and phagocytosed tau fragments by co-IP and fluorescence imaging. Moreover, they assessed how Tau feeding and PQBP1 knockout/knockdown impact primary microglial activation through NF-kB nuclei translocation, luciferase reporter, TNF α mRNA, etc. Finally, they showed in vivo data whereas conditional PQBP1 KO in microglia seems to impair microglia

activation without interfering with Tau uptake by other cells. This is the first report that attempts to link PQBP1-cGAS-STING signaling with Tau binding directly. While the overall story is novel, I have some concerns and advice for additional experiments that may consolidate these findings.

Major concerns:

1. The authors proposed that PQBP1 directly recognizes extrinsic Tau phagocytosed by microglia and increases microglial activation through the cGAS-STING pathway. However, I am not sure how much does PQBP1-Tau interaction contribute to microglial activation beyond PQBP1 interactions with other polyQ proteins.

>>> We appreciate the reviewer for suggesting an important viewpoint to evaluate our work in the whole frame of neurodegenerative diseases including polyQ diseases and tauopathy. As the reviewer might know, PQBP1 interacts with multiple polyQ diseases proteins such as Ataxin-1 and Huntingtin (Waragai et al, Hum Mol Genet 1999; Okazawa et al, Neuron 2002). Most of such interactions with polyQ sequences are mediated by intrinsically disordered C-terminal domain (CTD) of PQBP1.

On the other hand, PQBP1 has another domain for protein-protein interaction, i.e. WW domain (WWD) that recognizes proline-rich sequences (please refer to many works of Prof Marius Sudol). This applies to the specific case of interaction between PQBP1 and Tau, and as shown in Figure 1, WW domain of PQBP1 interacts with Proline sequences of Tau.

The reviewer might consider that polyQ proteins taken up by microglia also trigger PQBP1-cGAS-STING signaling to activate microglia, which is a very interesting idea and I by myself completely agree with this hypothesis. However, interaction domain is different as described above, and the proof for the cases of polyQ proteins needs a substantial amount of data that should be presented in two or three papers for each polyQ proteins.

Data in this manuscript showed that Tau 4R bind less to PQBP1 compared with 3R and induced less TNF α , which is counterintuitive as the tauopathy in HD pathology is associated with 4R repeats. Do the authors have an explanation on this? In other words, the authors show that PQBP1 can bind Tau, but this does not demonstrate that PQBP1 does bind *in vivo*.

>>> We again thank the reviewer for this critical comment. Regarding HD-associated Tau 4R, in our response to the comment #2 of this reviewer, we performed IP and IHC of R6/2 mice, and confirmed that *in vivo* PQBP1 does bind Tau in microglia in the brain (Supplementary Figure 2).

Especially about the comment of 4R, we would like to stress that Tau441 definitely interacts with PQBP1 at the affinity more than 60% of Tau 410. So it does not mean that PQBP1 cannot work in the HD pathology but that PQBP1 would be able to work via interaction with Tau 441 in microglia activation also of HD.

In addition, our new experiments to evaluate the effect of Tau species on neuronal death and memory functions *in vivo* in mouse (new Figure 9, 10, Supplementary Figure 8) revealed that Tau 4R, especially when it was polymerized, induced neuronal death and impaired memory rather independently of PQBP1, which seem mediated by direct toxicity to neurons judging from additional our new data.

2. The *in vivo* model is based on a Tau injection model where overall microglial activation was not impressive (<8% according to figure 5b, middle panels). The authors may utilize a disease mouse model to improve the results of this interaction *in vivo*. A direct evidence to show the PQBP1-Tau interaction *in vivo* occurs in a Huntington's disease model such as R2/6 mice might help.

>>> Regarding 8% of microglial activation in the middle graph of Figure 5b, the presentation of data had been incomplete.

We showed the percentage of activated microglia (=nuclear NFkB-positive microglia) among all microglia.

But it should be the percentage of activated microglia among Tau-incorporating microglia. Therefore, we changed the graph with more appropriate values, and now the graph shows that more than 90% of microglia were activated by Tau incorporation (Figure 5b, now moved to lower panels).

We also followed the advice of the reviewer and examined PQBP1-Tau interaction in vivo by using R6/2 mice (Supplementary Figure 2).

3. Recently, LRP1 expressed by microglia has been demonstrated as a dominant Tau receptor. Since results in this manuscript suggest that PQBP1-cGAS-STING contributes to microglial activation after Tau uptake. I would recommend the authors to include LRP1-KO in this study to further emphasize the sequential intervention of these pathways in Tau-uptake and microglia activation.

>>> We performed requested experiments using LRP1-KO microglia in parallel with TREM2-KO microglia (Figure 8). NFkB activation was monitored by NFkB cis-element reporter that expresses RFP fluorescent protein (Figure 8b,c). Some background signals are due to leaky expression of RFP from the lentivirus reporter. We also double-checked the combined effects of membrane receptor LRP1/TREM2 and cytoplasmic receptor PQBP1 on NFkB activity by luciferase assay (Figure 8b, d). These results (Figure 8) revealed genetic interaction and functional relationship between membrane receptor of Tau (LRP1 or TREM2) and cytoplasmic receptor of Tau (PQBP1) (Figure 8e).

4. With current data, the direct physical interaction between PQBP1 and Tau is not conclusively demonstrated. The authors could have utilized their NMR data to introduce some mutations into Tau fragments to block the interaction between PQBP1 and Tau.

>>> Due to some troubles of NMR we could not perform NMR analysis with Tau mutants. Corona virus situation also influenced the delay. Instead we performed SPR with Tau mutants possessing mutations at proline residues used for interaction with PQBP1 (Figure 1b). The results showed obvious decrease in their affinities to PQBP1.

Then, they could add this experimental group as negative control to show the change in co-localization and microglial response in vitro and in vivo.

>>> Yes, we did employed PQBP1-binding mutants of Tau, and observed their co-localization with PQBP1 or cGAS and examined their microglial responses in vitro and in vivo. In Figure 2a and c, we revealed the effect of proline mutation of Tau on co-localization of PQBP1 and Tau. In Figure 3b and c, we revealed the effect of proline mutation of Tau on microglial response in vitro. In Figure 7, we revealed the effect of proline mutation of Tau on microglial response in vivo. In Figure 9, we tested the effect of proline mutation of Tau on neuronal death in vivo. Finally in Figure 10, we revealed their effect on mouse memory functions.

5. Could the authors show the Kd values determined by their SPR assay? In figure 1B, the concentration of the proteins used seems too high.

>>> We showed the Kd values in Figure 1b. PQBP1 bind to normal Tau at 10^{-8} (10 to the power of minus 8) order of Kd, while it binds to proline-mutant of Tau at 10^{-5} (10 to the power of minus 5) order. Kd at 10^{-8} order will be sufficient for protein-protein interaction in the cytoplasm at the concentrated foci.

6. In figure 2, could the authors also show quantitative analysis as what they did for figure 3c?

>>> Old Figure 2 is now new Figure 3 in revised version. We added quantitative analysis (Figure 3b, graph) like Figure 4c (=pld Figure 3c).

7. While the water maze results in figure 5d demonstrated that conditional PQBP1-KO in microglia rescues cognition defects caused by Tau injection, it was also clear that such KO in microglia causes massive Tau uptake by non-microglia cells (figure 5b, bottom panels). Given that PQBP1 is not exclusively expressed in microglia in the CNS, did the authors evaluate the identity and fate of those cells that uptake Tau in a large amount?

>>> We appreciate the comment from the reviewer. We agree that non-microglia cells take up Tau in vivo. It is rather popular in the research field to investigate prionoid transmission of Tau, and basically the technique used in our study is similar to such experiments of other researchers.

As requested, we evaluated the fate of non-microglia cells that uptake Tau injected to the brain in a large amount (new Figure 9). First, we examined cell types that take up Tau, and found that the non-microglia cells were mostly neurons (new Figure 9a).

Our further analyses, though mostly as the repetition of other researchers' results, revealed that neurons taking up polymerized disease-mutant Tau (Tau 441-P301S) underwent cell death (Tunel-positive) (new Figure 9b, Ext Data Figure 8i). However, the Tau species (monomer/polymer Tau 410, monomer Tau 441) that we think important for microglia activation did not have a large impact on neuronal cell death (new Figure 9b, Ext Data Figure 8i). The similar result was obtained in primary culture neurons. Polymerized disease-mutant Tau caused neuronal cell death, but other species except Tau 441 polymer did not have such a direct toxicity to neurons (new Figure 9c, Ext Data Figure 8j).

The reviewer's comment led to clearer elucidation of two parallel pathways from different species of Tau to different types of cells (microglia vs neuron), which is very interesting. We really appreciate it, and we added one section in Discussion regarding this topic.

8. The assessment of downstream of PQBP1-cGAS-STING pathway in this study could be extended to type I interferon response such as IFN β , CXCL10, etc., not just NF-kB nuclear translocation.

>>> We have already analyzed IRF3-dependent ISG54 in addition to NFkB-dependent TNFa in our previous version, so we do not think that we only (just) investigated NFkB.

However, following the advice we did examine responses of IFN β and CXCL10, as additional examples of NFkB-dependent and IRF3-dependent genes, and revealed that these genes were changed similarly to TNFa and ISG54 (Figure 5e).

Reviewer #3 (Remarks to the Author):

The manuscript from Jin et al. examines the role of PQBP1 in Tau-mediated activation of microglia. The work begins with an evaluation of the Tau – PQBP1 interaction in microglia, and experiments are performed in primary mouse microglia, which is a physiologically appropriate system, and analytical methods and statistics are sound. The authors present data suggesting that the PQBP1 interaction with Tau occurs in the ER, which is where STING is also found, and show that the STING pathway is activated upon PQBP1 – Tau interaction in microglia. After documenting the necessity of PQBP1 for robust cGAS-STING activation in vitro, the study transitions to provocative in vivo experiments taking advantage of the authors' existing PQBP1 conditional knock-out mouse model. Using a co-injection paradigm of Tau monomer + tamoxifen to turn off PBQP-1 gene expression, they report impressive findings for read-outs including microglia activation, NF-kb nuclear translocation, cGAS-STING activation, and importantly, cognitive and memory function.

>>> We thank the reviewer very much for kind evaluation and critical suggestions.

While it seems unlikely PQBP1 is expressed in microglia solely to be a sensor for misfolded Tau or other CNS insults, my assessment is that this work is novel and could represent a substantive advance.

>>> Thank you very much for the deep insight about the function of PQBP1 and kind evaluation of the reviewer. We examined chemotaxis and phagocytosis function of PQBP1-depleted microglia (Figure 4e, f), as other functions of PQBP1 in microglia.

In addition, as we reported previously (Tanaka et al, Mol Psy 2018), PQBP1 regulates synapse functions, which is disturbed in AD pathology. We performed first set of investigations to analyze the effect of incorporated Tau on PQBP1-dependent neuronal damage (Figure 9, Supplementary Figure 8), while it was not remarkable. However, we believe that this issue should be investigated further in the future.

Issues with this work that the authors need to address are as follows:

1) The authors note that PQBP1 depletion reduces NF-kb nuclear translocation by 50%, suggesting that other pathways are also operational. Can the authors elaborate upon what other pathways might be relevant here? They imply that in AD, there is another pathway stemming from amyloid-beta sensing involving TREM2, but can the authors test if TREM2 modulation alters the response to Tau in PQBP1 depleted microglia?

>>> We appreciate the reviewer for the suggestion of TREM2 as the molecule mediating PQBP1-independent pathway for microglia activation in response to Tau. We added experiments for this question in new Figure 8. Together with LRP1, which reviewer #2 suggested as a Tau receptor, we generated TREM2-KO or LRP1-KO microglia with or without double KO of PQBP1 (tamoxifen-induced PQBP1-KO), and investigated their responses to Tau 410 monomer.

Consequently, the genetic interaction analysis indicated no additive relationship between LRP1 and PQBP1 but an additive relationship between TREM2 and PQBP1, bringing us to a scheme of two pathways, LRP1-PQBP1 and TREM2-ERK/PI3K (new Figure 8).

2) I find it curious that the microglia response to 3R-Tau and 4R-Tau is comparable in your experiments. Did you also consider evaluating Tau carrying FTD-linked mutations or other relevant functional variants?

>>> We appreciate very much the critical comment from the reviewer to strengthen our work. As the review kindly and correctly understands, 4R triggers microglial activation (Figure 3a, 3b, 3c, 4b, 4c, 4d, 5a, 5b, 5c, 5d). Following the advice, we tested the FTL-linked mutation of Tau (Tau P301S) as well as binding mutants (Tau P179A, Tau P216A, and Tau P179A/P216A) in binding to PQBP1 in vitro (Figure 1, Supplementary Figure 8), microglia activation in culture (Figure 3, Supplementary Figure 8), and microglia activation in vivo (Figure 7, Supplementary Figure 8).

3) One overriding question is the normal function of PQBP1 in microglia. Have you carefully analyzed other functions of microglia depleted of PQBP1? Are chemotaxis and phagocytosis intact?

>>> We tested chemotaxis and phagocytosis of PQBP1-deficient microglia (Figure 4e, f) and their functions were normal as expected.

4) Do mice depleted of PQBP1 in microglia develop any neurological phenotypes? In other words, should we be thinking of PQBP1 as a target for dosage reduction in Alzheimer's disease? I would like the authors to discuss whether or not PQBP1 is a viable therapeutic target in the final section of the paper.

>>> We have observed Tamoxifen-cKO mice for six months, but they did not show any abnormality in their motor activity, eating, and other appearance and

behavior, although they were kept in the SPF condition. Following the advice of the reviewer, we added one section at the end of Discussion in our manuscript, to discuss about whether PQBP1 in microglia could be a therapeutic target.

REVIEWERS' COMMENTS

Reviewer #1 (Remarks to the Author):

I apologize for the late review due to some health issues. As in my initial review, I find the work sound and of importance. The authors addressed my concern. I highly recommend the publication of the manuscript in Nature Communication.

Reviewer #2 (Remarks to the Author):

In this revised manuscript, Meihua Jin et al. have adequately responded to my previous comments and provided additional data addressing all concerns. I am very enthusiastic and satisfied by the comprehensive data and the intriguing story presented in this version. I think the manuscript is suitable for publication except for some minor edits.

1. It is nice to show microglia activation (e.g., nuclear NF- κ B) exclusively in Tau-incorporating microglia (New figure 6b). However, I would recommend the authors to keep the original figure in supplementary figures since that data suggested a relatively low microglia activation in response to Tau injection.

2. I really appreciate the authors new model and believe it is of significance to present it with the data. But perhaps it would be better to combine figure 9d and 10b into one complete scheme. Also, it is possible that mono-Tau taken up by microglia contributes to the intracellular polymerization of new tau aggregates, which may help explain why microglia depletion in Tau mouse models turned to be protective.

3. Some figure legends for the new panels need to be added.

Reviewer #3 (Remarks to the Author):

The revised manuscript is much improved with addition of a number of experiments and new results which strengthen paper. I am therefore very satisfied with authors' response to my concerns.

I did find one issue. Figure 9d is a very confusing model, and does not really add much to the manuscript. The fact that there is no legend for panel d of Figure 9 may have contributed to this problem, but even so, I suggest that this model be clarified or removed from the paper.

REVIEWERS' COMMENTS

Reviewer #1 (Remarks to the Author):

I apologize for the late review due to some health issues. As in my initial review, I find the work sound and of importance. The authors addressed my concern.

I highly recommend the publication of the manuscript in Nature Communication.

>>> Thank you very much for the kind evaluation. We really appreciate thoughtful comments from reviewer #1.

Reviewer #2 (Remarks to the Author):

In this revised manuscript, Meihua Jin et al. have adequately responded to my previous comments and provided additional data addressing all concerns. I am very enthusiastic and satisfied by the comprehensive data and the intriguing story presented in this version. I think the manuscript is suitable for publication except for some minor edits.

>>> Thank you very much for the kind evaluation. We really appreciate great efforts of reviewer #2 for improving our manuscript.

1. It is nice to show microglia activation (e.g., nuclear NF- κ B) exclusively in Tau-incorporating microglia (New figure 6b). However, I would recommend the authors to keep the original figure in

supplementary figures since that data suggested a relatively low microglia activation in response to Tau injection.

>>> Following the advice, we kept the original figure in new Supplementary Figure 7. However, we would like to stress that Figure 6b shows the ratio of nuclear NF- κ B-positive microglia among Tau-positive microglia, while new Supplementary Figure 7 shows the ratio of nuclear NF- κ B-positive microglia among *TOTAL* microglia, most of which do not uptake Tau. Therefore, microglia activation is high when Tau is incorporated to microglia.

2. I really appreciate the authors new model and believe it is of significance to present it with the data. But perhaps it would be better to combine figure 9d and 10b into one complete scheme.

>>> We appreciate this suggestion from reviewer #2. Figure 9d introduced two pathways (direct pathway and indirect pathway via microglia) from extracellular Tau to neuronal damage. Figure 10b shifts the viewpoint to the microglia \rightarrow neuron and neuron \rightarrow microglia dual pathways. Therefore, each scheme has a specific claim (explanation) for each Figure, and integration of the two panels may prevent step-by-step understanding of readers. We actually asked PIs in other laboratories, and they said it becomes too complex and difficult to understand the story if the two panels are combined.

Also, it is possible that mono-Tau taken up by microglia contributes to the intracellular polymerization of new tau aggregates, which may help explain why microglia depletion in Tau mouse models turned to be protective.

>>> We appreciate this interesting hypothesis. We incorporated the idea to Figure 9d by referring a paper that supports this idea (Asai et

al, Nat Neurosci 2015) together with a paper revealing different aspects (Leyns et al, Nat Neurosci 2019).

3. Some figure legends for the new panels need to be added.

>>> We added legend to Figure 9d.

Reviewer #3 (Remarks to the Author):

The revised manuscript is much improved with addition of a number of experiments and new results which strengthen paper. I am therefore very satisfied with authors' response to my concerns.

>>> Thank you very much for the kind evaluation. We really appreciate kind comments from reviewer #3.

I did find one issue. Figure 9d is a very confusing model, and does not really add much to the manuscript.

The fact that there is no legend for panel d of Figure 9 may have contributed to this problem, but even so, I suggest that this model be clarified or removed from the paper.

>>> We suspect the reviewer #3 's confusion might come from the mouse background of our model. We did not use Alzheimer's disease model mice for PQBP1-cKO. From our data, Tau injection increased total number of microglia, but did not increase the ratio of disease-associated microglia (DAM) among total microglia. Also from the data of Supplementary Figure 8, activated microglia in our experiments are LPL-negative (non-DAM in Figure 9d). So if

reviewer #3 assumes LPL-positive microglia (DAM) around A β plaques, our scheme might be confusing.

On the other hand, reviewer #2 is highly interested in Figure 9d, and proposed that this might explain Tau propagation via microglia. He/She suggests that Tau might be aggregated in microglia. We suspect that reviewer #2 probably assumes non-DAM. Of course as shown in the paper by Keren-Shaul et al in Cell 2017, non-DAM exist in remote regions from amyloid plaques and a few non-DAM exist even around plaques in AD pathology.

To reconcile these different views on microglia in AD pathology, we have chosen to clarify this model rather than to discard it. We included DAM in Figure 9d, for clarification. We also added text sentences and figure legend for explaining Figure 9d.